# HIF1α inhibition facilitates Leflunomide-AHR-CRP signaling to attenuate bone erosion in CRP-aberrant rheumatoid arthritis

Chao Liang [1,2,3,8,9], Jie Li[4,8], Cheng Lu[1,2,3,5,8], Duoli Xie[1,2,3], Jin Liu[1,2,3], Chuanxin Zhong[1,2,3], Xiaohao Wu[1,2,3], Rongchen Dai[1,2,3], Huarui Zhang[1,2,3], Daogang Guan[1,2,3], Baosheng Guo[1,2,3], Bing He[1,2,3], Fangfei Li[1,2,3], Xiaojuan He[1,2,3,5], Wandong Zhang[6], Bao-Ting Zhang[4,9], Ge Zhang[1,2,3,9] & Aiping Lu[1,2,3,7,9]

Rheumatoid arthritis (RA) is a chronic inflammatory disorder characterized by progressive bone erosion. Leflunomide is originally developed to suppress inflammation via its metabolite A77 1726 to attenuate bone erosion. However, distinctive responsiveness to Leflunomide is observed among RA individuals. Here we show that Leflunomide exerts immunosuppression but limited efficacy in RA individuals distinguished by higher serum C-reactive protein (CRP$^{Higher}$, CRP$^H$), whereas the others with satisfactory responsiveness to Leflunomide show lower CRP (CRP$^{Lower}$, CRP$^L$). CRP inhibition decreases bone erosion in arthritic rats. Besides the immunomodulation via A77 1726, Leflunomide itself induces AHR-ARNT interaction to inhibit hepatic CRP production and attenuate bone erosion in CRP$^L$ arthritic rats. Nevertheless, high CRP in CRP$^H$ rats upregulates HIF1α, which competes with AHR for ARNT association and interferes Leflunomide-AHR-CRP signaling. Hepatocyte-specific *HIF1α* deletion or a HIF1α inhibitor Acriflavine re-activates Leflunomide-AHR-CRP signaling to inhibit bone erosion. This study presents a precision medicine-based therapeutic strategy for RA.

[1] Law Sau Fai Institute for Advancing Translational Medicine in Bone and Joint Diseases, School of Chinese Medicine, Hong Kong Baptist University, 999077 Hong Kong SAR, China. [2] Institute of Integrated Bioinfomedicine and Translational Science, School of Chinese Medicine, Hong Kong Baptist University, 999077 Hong Kong SAR, China. [3] Institute of Precision Medicine and Innovative Drug Discovery, HKBU Institute for Research and Continuing Education, 518000 Shenzhen, China. [4] Faculty of Medicine, School of Chinese Medicine, Chinese University of Hong Kong, 999077 Hong Kong SAR, China. [5] Institute of Basic Research in Clinical Medicine, China Academy of Chinese Medical Sciences, 100700 Beijing, China. [6] The First Affiliated Hospital of Anhui University of Chinese Medicine, 230031 Hefei, China. [7] Institute of Arthritis Research, Shanghai Academy of Chinese Medical Sciences, Guanghua Integrative Medicine Hospital/Shanghai University of TCM, 200032 Shanghai, China. [8] These authors contributed equally: Chao Liang, Jie Li, Cheng Lu. [9] These authors jointly supervised this work: Chao Liang, Bao-Ting Zhang, Ge Zhang, Aiping Lu. Correspondence and requests for materials should be addressed to G.Z. (email: zhangge@hkbu.edu.hk)

 1

Rheumatoid arthritis (RA) is an autoimmune disease characterized by inflammatory synovial hyperplasia, stiffness, progressive bone erosion and systemic features. Progressive bone erosion is a prominent determinant of functional impairment of joints in RA[1]. Currently, a range of treatment options including conventional and biologic antirheumatic drugs are available[2,3]. However, none of them is universally effective[4,5]. Understanding the mechanisms underlying the non-responses of these drugs will provide new insight into development of precision medicine-based therapeutic strategy for RA[5].

During RA development, multiple immune cells and cytokines are organized within a hierarchical regulatory network that favors osteoclast-mediated bone erosion[6]. Leflunomide has been widely used in RA treatment and the well-known mechanism is immunomodulation via its metabolite A77 1726[7]. Briefly, Leflunomide is converted to A77 1726 in intestinal wall and liver after oral administration. A77 1726 inhibits mitochondrial dihydroorotate dehydrogenase (DHODH) required for proliferation of immune cells, resulting in attenuation of inflammation-induced bone erosion[8]. However, only 40–50% RA patients receiving Leflunomide meet 20% reduction in disease activity according to American College of Rheumatology response criteria[9].

In this study, we classified RA patients treated with Leflunomide based on presence of progressive bone erosion (PBE-positive, PBE+) or not (PBE-negative, PBE−). Interestingly, Leflunomide significantly inhibited inflammation in both PBE− and PBE+ patients, suggesting the limited efficacy of Leflunomide in attenuating bone erosion in PBE+ patients were not coupled with its immunomodulatory action. The PBE+ patients had higher serum C-reactive protein (CRP$^{Higher}$, CRP$^H$), whereas the PBE- patients showed relatively lower serum CRP (CRP$^{Lower}$, CRP$^L$). Studies have demonstrated that CRP of hepatic origin is involved in osteoclastogenesis[10,11]. We found that high CRP in CRP$^H$ patients was positively correlated with increased bone resorption. Data from collagen-induced-arthritis (CIA) animal models was consistent with the above findings from RA patients.

Inhibition of CRP by anti-CRP antibodies or liver-targeted siRNA delivery systems[12] led to attenuated bone erosion in CRP$^H$ CIA rats. CRP expression is under the negative regulation of aryl hydrocarbon receptor (AHR) genomic signaling[13], which requires binding with AHR nuclear translocator (ARNT) for activation upon stimulation of agonists[14]. Accumulating evidences suggest that Leflunomide could be an AHR agonist[15]. We showed that Leflunomide, rather than A77 1726, significantly induced binding of ARNT with AHR to inhibit CRP expression in normal hepatocytes, but not in hepatocytes overexpressing CRP in vitro. Besides the well-known immunomodulatory action via A77 1726, Leflunomide itself activated AHR-CRP signaling and then attenuated bone erosion in CRP$^L$ CIA rats but not in CRP$^H$ CIA rats.

Hypoxia-inducible factor 1α (HIF1α) has been reported to compete with AHR for ARNT association[16]. We found that CRP upregulated HIF1α expression, which competed with AHR for ARNT association and interfered Leflunomide-AHR-CRP signaling in CRP$^H$ CIA rats. Knockdown of HIF1α re-activated Leflunomide-AHR-CRP signaling in hepatocytes overexpressing CRP in vitro. Hepatocyte-specific deletion of HIF1α improved the Leflunomide-AHR-CRP signaling to inhibit bone erosion in CRP$^H$ CIA mice. Acriflavine (ACF), a FDA-approved drug, has been reported as a selective inhibitor targeting HIF1α[17]. We showed that ACF decreased binding of ARNT with HIF1α and facilitated Leflunomide activating AHR to inhibit CRP production and attenuate bone erosion in CRP$^H$ CIA rats with no obvious toxicity.

In summary, this study reveals that CRP-HIF1α signaling axis is responsible for the limited efficacy of Leflunomide in CRP$^H$ RA. On the basis of this finding, we develop a precision medicine-based therapeutic strategy for CRP$^H$ RA, i.e., the combination of Leflunomide and ACF.

## Results

**Limited efficacy of Leflunomide in CRP$^H$ RA patients.** We reviewed radiographic data of 250 RA patients treated with Leflunomide (Supplementary Table 1). Leflunomide significantly attenuated progressive bone erosion in 130 RA patients (PBE−) but showed limited efficacy in the rest 120 RA patients (PBE+) (Fig. 1a, b). However, inhibition of DHODH activity and proliferation of immune cells (T and B lymphocytes and macrophages) were comparable between PBE− and PBE+ patients. Cytokines produced by immune cells and inflammatory synovial fibroblasts including Interleukin-17 (IL-17), Interleukin-6 (IL-6), and receptor activator of nuclear factor kappa-B ligand (RANKL)[18] also showed no difference between the two RA groups (Fig. 1c and Supplementary Fig. 1a). We determined the associations between PBE + patients and serum baseline blood indicators including rheumatoid factors (IGM, IGG and IGA), CRP, anti-cyclic citrullinated peptide (anti-CCP) antibody and erythrocyte sedimentation rate (ESR)[19]. Serum CRP showed high specificity and sensitivity for PBE+ RA patients and the diagnostic accuracy was above 92% (Fig. 1d, Supplementary Fig. 1b and Supplementary Table 2). The PBE+ patients demonstrated higher levels of serum baseline CRP (CRP$^H$) and a bone resorption marker (tartrate-resistant acid phosphatase 5b, TRAP5b)[20], whereas the PBE− patients showed relatively lower CRP (CRP$^L$) and TRAP5b (Fig. 1e). During Leflunomide treatment, serum levels of both CRP and TRAP5b were significantly inhibited in CRP$^L$ but not in CRP$^H$ patients (Fig. 1f). Serum CRP, rather than other indicators, was positively associated with TRAP5b in CRP$^H$ RA patients (Fig. 1g and Supplementary Fig. 1c). Role of CRP in osteoclastogenesis are conformation- and RANKL-dependent. Circulating native CRP is composed of five identical subunits and dissociates into the monomeric conformation upon entering local lesions[21]. Monomeric CRP promotes osteoclast differentiation in the absence of RANKL but inhibits RANKL-induced osteoclastic differentiation by neutralizing RANKL[11]. We quantified the baseline monomeric CRP and RANKL in synovial fluid from RA patients. Molar concentration of monomeric CRP was over 10,000-fold of RANKL in both CRP$^L$ and CRP$^H$ RA patients (Supplementary Fig. 1d), suggesting that the monomeric CRP in the two groups of RA patients was enough to neutralizing RANKL and the redundant free monomeric CRP would dominate osteoclastic activities in RA.

**Limited efficacy of Leflunomide in CRP$^H$ CIA rats.** We established CIA rat models and administered the CIA rats with Leflunomide for 28 days (Supplementary Fig. 2a). The CIA rats were also classified into PBE+ and PBE− subgroups based on microCT analysis (Fig. 2a). We observed the progressively deteriorated bone microarchitecture and decreased bone mass in PBE+ rats rather than in PBE− rats after Leflunomide treatment (Fig. 2a and Supplementary Fig. 2b). Inhibition of DHODH activity and proliferation of immune cells (T and B lymphocytes and macrophages), as well as decrease of cytokines (IL-17, IL-6 and RANKL) were comparable between PBE− and PBE+ rats (Fig. 2b and Supplementary Fig. 2c). Inflammatory synovial hyperplasia was inhibited by Leflunomide in both PBE− and PBE+ rats (Supplementary Fig. 2d). The PBE+ rats demonstrated higher serum baseline CRP (CRP$^H$, 91.24 ± 10.12 mg L$^{-1}$) and TRAP5b, whereas the PBE− patients showed relatively lower CRP (CRP$^L$, 52.45 ± 7.54 mg L$^{-1}$) and TRAP5b (Fig. 2c). Leflunomide decreased serum CRP and TRAP5b levels and inhibited

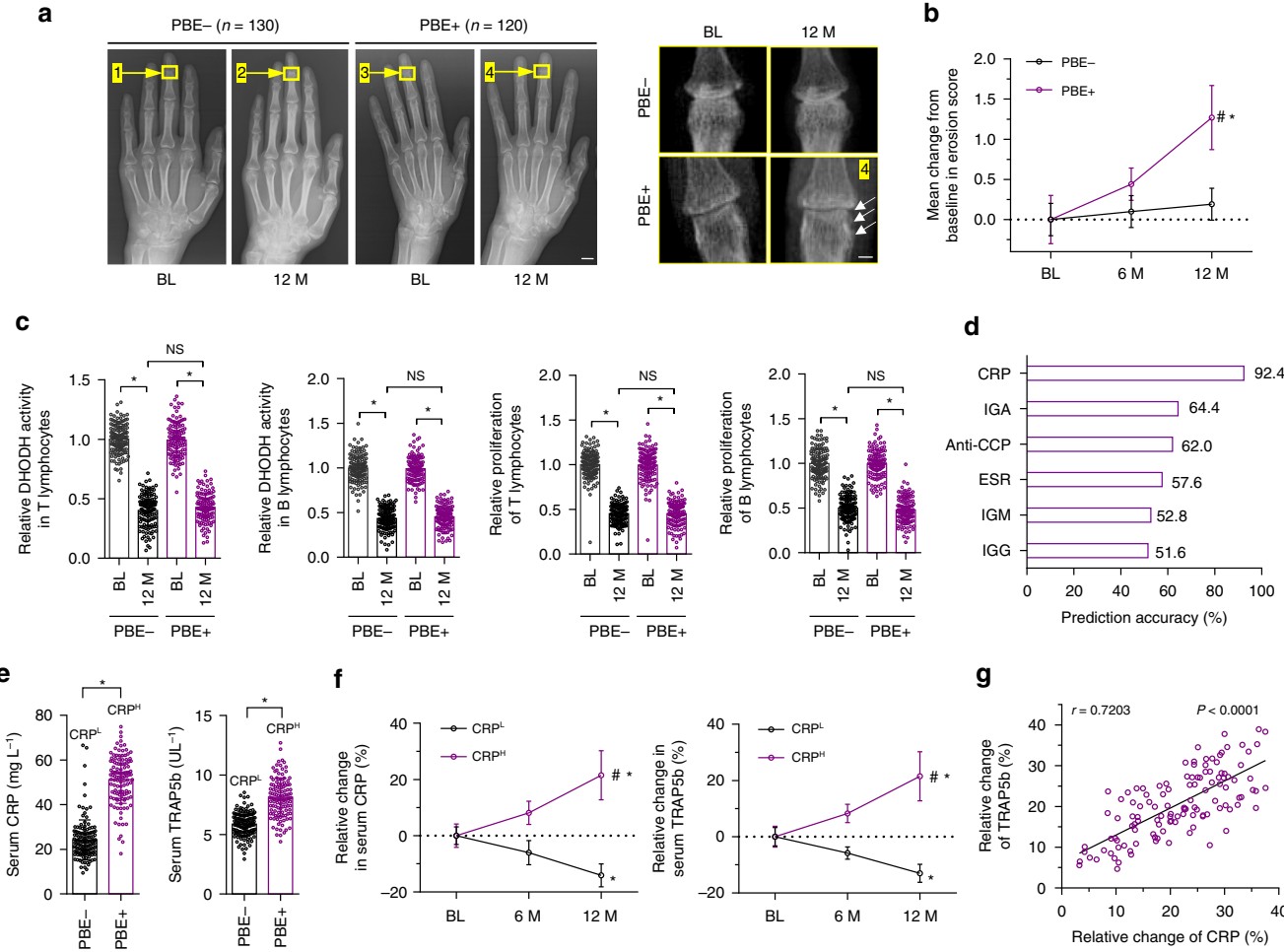

**Fig. 1** Differential responsiveness to Leflunomide among RA patients. **a** The representative hand X-ray radiographs (left) and enlarged images of interphalangeal joints (right) showing bone erosion in progressive bone erosion-positive (PBE+, indicated by white arrows, $n = 120$) and progressive bone erosion-negative (PBE−, $n = 130$) RA patients before (baseline, BL) and after (12 months, 12 M) Leflunomide treatment. Yellow arrows and boxes indicated the interphalangeal joints. Scale bars, 1.0 cm (left) and 5.0 mm (right). **b** Change of erosion score from baseline (BL). *P*-value was generated by repeated-measures analysis of variance (ANOVA) with a post hoc test. *$P < 0.05$ for 12 M versus BL in PBE+ and PBE−, #$P < 0.05$ for PBE+ versus PBE− at 12 M. **c** Relative levels of DHODH activity and proliferation of B and T lymphocytes in synovial fluid normalized to the corresponding baseline (BL) in PBE+ ($n = 120$) and PBE− ($n = 130$) patients. *$P < 0.05$ as determined by repeated-measures ANOVA with a post hoc test, NS: no significance. **d** Binary logistic regression of the PBE+ ($n = 120$) patients using blood indicators (IGM, IGG, IGA, CRP, ESR and anti-CCP). **e** The serum baseline CRP and TRAP5b levels in PBE+ ($n = 120$) and PBE− ($n = 130$) patients. CRP$^L$: relatively lower CRP level (CRP$^{Lower}$, 24.24 ± 8.94 mg L$^{-1}$). CRP$^H$: higher CRP level (CRP$^{Higher}$, 51.17 ± 10.60 mg L$^{-1}$). *$P < 0.05$ as determined by two-sided *t*-test. **f** Relative changes of serum CRP (left) and TRAP5b (right) from the corresponding baseline (BL) in CRP$^H$ ($n = 120$) and CRP$^L$ ($n = 130$) patients. *$P < 0.05$ for 12 M versus BL in CRP$^H$ and CRP$^L$, # $P < 0.05$ for CRP$^H$ versus CRP$^L$ at 12 M, as determined by repeated-measures ANOVA with a post hoc test. **g** Correlation between the relative change of serum CRP with TRAP5b from the corresponding baseline in CRP$^H$ RA patients ($n = 120$) as determined by Pearson's correlation. Source data are provided as a Source Data file

bone resorption and osteoclastic activity in CRP$^L$ rats but not in CRP$^H$ rats (Fig. 2d, e, Supplementary Fig. 2d). We investigated the mechanism underlying the distinctive CRP levels between the two groups of RA individuals. IL-6 axis was the chief inducer of CRP expression in hepatocytes[22]. We found that serum IL-6 were comparable between CRP$^L$ and CRP$^H$ rats (Supplementary Fig. 2c), while IL-6 receptor (IL-6R) in liver was differentially expressed between the two groups (Supplementary Fig. 2e).

**Attenuation of bone erosion by CRP inhibition in CIA rats.** The CRP$^H$ CIA rats were intra-articularly injected with PBS (vehicle), IgG controls or anti-CRP antibodies (Supplementary Fig. 3a). The anti-CRP antibodies attenuated bone erosion and bone resorption and prevented bone loss in CRP$^H$ rats when compared to IgG or PBS (Supplementary Fig. 3b–d). We also

used an RNA interference-based strategy to inhibit hepatic CRP expression in CRP$^H$ CIA rats. Lipid nanoparticles (LNPs) has been proven as liver-targeted delivery systems for siRNAs in vivo[23,24]. The CRP$^H$ CIA rats were intravenously administrated with PBS (vehicle), LNPs, LNPs encapsulating negative control siRNA (LNPs-NC siRNA) or LNPs encapsulating *CRP* siRNA (LNPs-CRP siRNA) (Supplementary Fig. 3e). About 90% of the administered dose of LNPs-siRNA was found in liver (Supplementary Table 3). LNPs-*CRP* siRNA lowered CRP expression in hepatocytes, attenuated bone erosion and bone resorption and prevented bone loss in CRP$^H$ rats when compared to PBS, LNPs or LNPs-NC siRNA (Supplementary Fig. 3f–i). We determined whether CRP knockdown by LNPs-siRNA could abrogate the differential responsiveness to Leflunomide between CRP$^H$ and CRP$^L$ CIA rats. Our results showed that Leflunomide in combination with LNPs-siRNA effectively attenuated bone

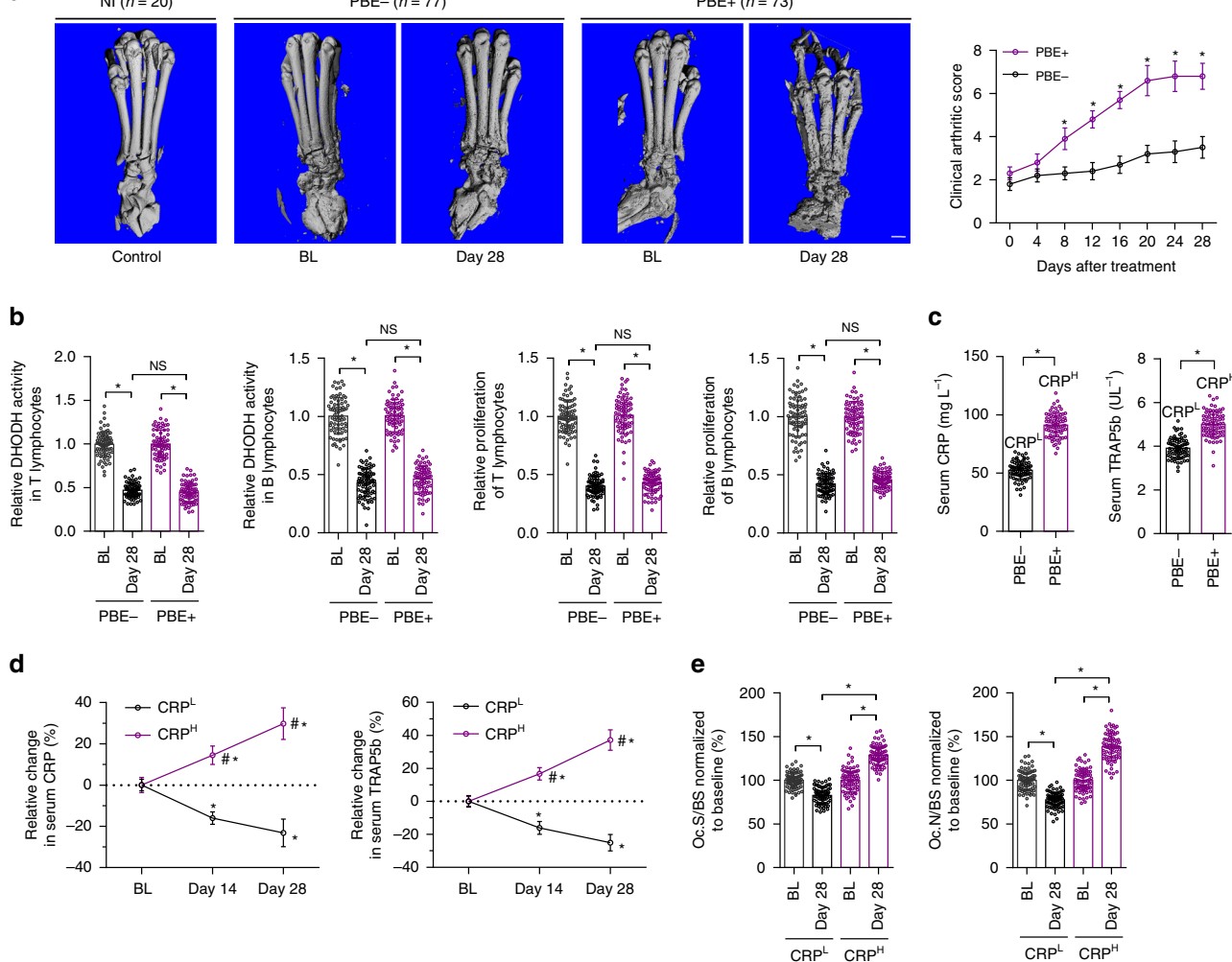

**Fig. 2** Differential responsiveness to Leflunomide among CIA rats. **a** The representative three-dimensional microCT images and clinical arthritic scores of the hind paws from PBE+ (n = 73) and PBE− (n = 77) CIA rats before (baseline, BL) and after Leflunomide treatment. Non-immunized (NI) rats (n = 20) were used as controls of the CIA rats. Scale bar, 5.0 mm. *P < 0.05 as determined by repeated-measures ANOVA with a post hoc test. **b** Relative levels of DHODH activity and proliferation of T and B lymphocytes in synovial fluid normalized to the corresponding baseline in PBE+ (n = 73) and PBE− (n = 77) CIA rats. *P < 0.05 as determined by one-way ANOVA with a post hoc test, NS: no significance. **c** The serum baseline CRP and TRAP5b levels in PBE− (n = 77) and PBE+ (n = 73) rats. CRP$^L$: relatively lower CRP level (CRP$^{Lower}$, 52.45 ± 7.54 mg L$^{−1}$). CRP$^H$: higher CRP level (CRP$^{Higher}$,91.24 ± 10.12 mg L$^{−1}$). *P < 0.05 as determined by two-sided t-test. **d** Relative changes of serum CRP (left) and TRAP5b (right) from the corresponding baseline in CRP$^H$ rats (n = 73) and CRP$^L$ (n = 77) rats. *P < 0.05 for day 14 or day 28 versus BL in CRP$^L$ and CRP$^H$, #P < 0.05 for CRP$^H$ versus CRP$^L$ at day 14 and day 28, as determined by repeated-measures ANOVA with a post hoc test. **e** Bone resorption parameters including osteoclast surface per bone surface (Oc.S/BS) and osteoclast number per bone surface (Oc.N/BS) in CRP$^H$ (n = 73) and CRP$^L$ (n = 77) rats after normalization to the corresponding baseline. *P < 0.05 as determined by one-way ANOVA with a post-hoc test. Source data are provided as a Source Data file

erosion in CRP$^H$ CIA rats. The therapeutic efficacy of the combination in CRP$^H$ CIA rats was equivalent with that of Leflunomide in CRP$^L$ CIA rats (Supplementary Fig. 4a–d). We also examined the effects of anti-CRP antibodies in CRP$^L$ CIA rats. The anti-CRP antibodies partially mimic the therapeutic efficacy of Leflunomide in attenuating bone erosion (Supplementary Fig. 4e–h).

**Direct interaction between Leflunomide and AHR.** AHR genomic signaling plays an essential role in controlling hepatic CRP production[13]. Upon stimulation with agonists, AHR binds to ARNT to inhibit CRP expression[13,14,25]. Accumulating evidence suggests that Leflunomide, rather than its metabolite A77 1726, could be an AHR agonist[26]. To confirm the direct interaction between Leflunomide and AHR, we incubated Flag-tagged

AHR (Flag-AHR) with DMSO (vehicle), Leflunomide and A77 1726, respectively. After pull-down assay, we detected that AHR interacted with Leflunomide rather than A77 1726 (Fig. 3a). To identify the key residues involved in Leflunomide-AHR interaction, we conducted homology modeling of AHR PAS-A and PAS-B domains, which are required for dimerization with ARNT and interaction with agonists[14]. The optimal structure was chosen for molecular docking with Leflunomide (Supplementary Fig. 5a). Six binding modes were predicted (Supplementary Fig. 5b). Key residues in the best two binding modes including H291, K303, and V381 were chosen for mutation (Supplementary Table 4). Mutation of either H291 or K303 decreased the binding between Leflunomide and AHR, whereas mutation of V381 could not affect Leflunomide-AHR interaction. Mutation of both H291 and K303 resulted in a more significant decrease in Leflunomide-AHR interaction (Fig. 3b). We further conducted drug affinity

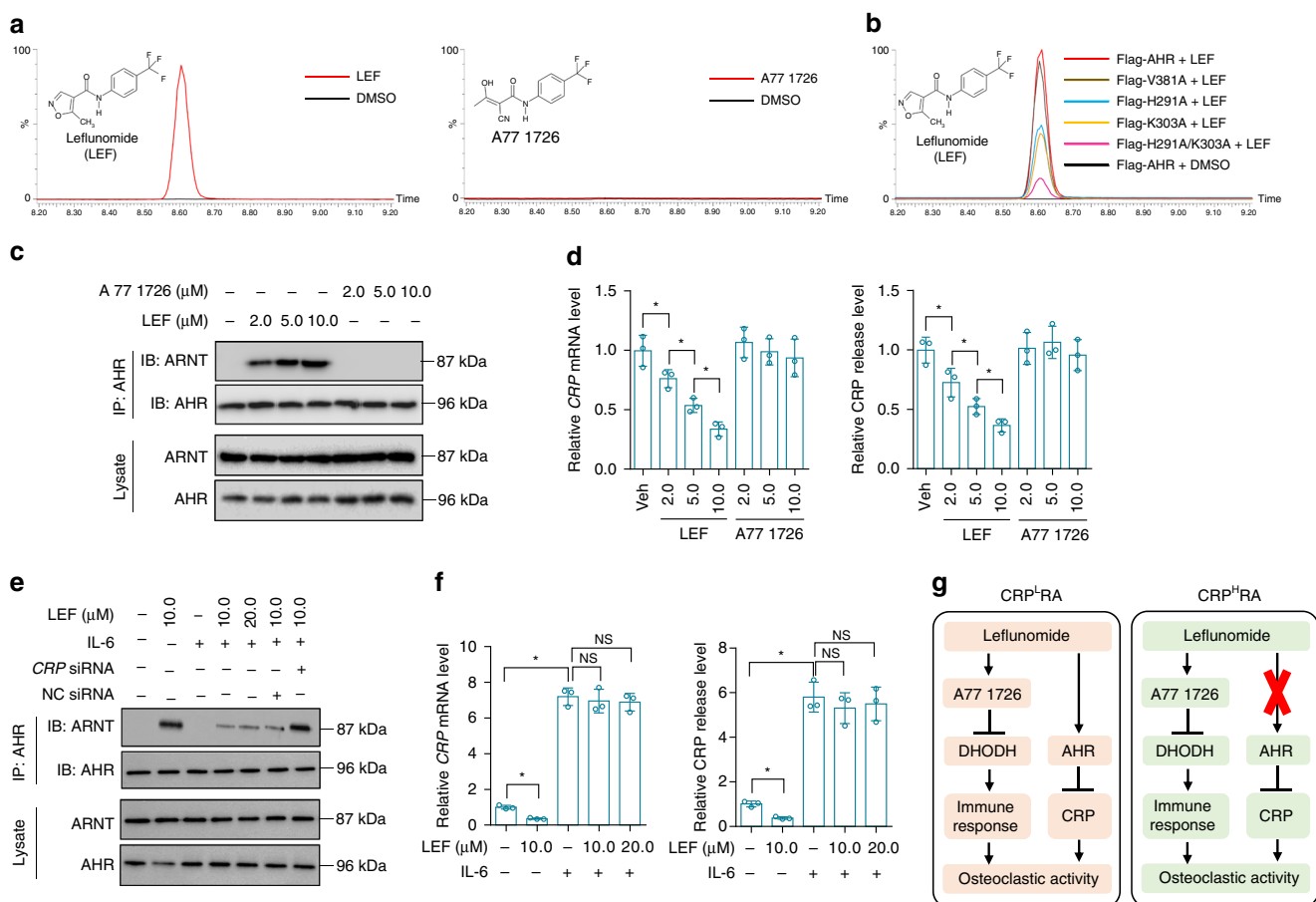

**Fig. 3** Effects of Leflunomide on AHR genomic signaling and CRP expression in vitro. **a** Pull-down assay for interaction between Flag-tagged AHR (Flag-AHR) and Leflunomide (LEF, left) or A77 1726 (right). Flag-AHR was incubated with Leflunomide, A77 1726 and vehicle (DMSO), respectively. Small molecules captured by Flag-AHR were determined by liquid chromatography-tandem mass spectrometry (LC-MS/MS). **b** Pull-down assay for interaction between AHR mutants and Leflunomide (LEF). Flag-AHR and AHR mutants (Flag-H291A, Flag-K303A, Flag-V381A and Flag-H291A/K303A) was incubated with Leflunomide or vehicle (DMSO). **c** Binding of ARNT with AHR determined by immunoprecipitation in human normal hepatocytes (THLE-2) incubated with vehicle (DMSO), Leflunomide (LEF) and A77 1726 at a series of concentrations, respectively. **d** Relative levels of *CRP* mRNA (left) and CRP release (right) in THLE-2 cells incubated with Leflunomide (LEF), A77 1726 and vehicle (Veh, DMSO), respectively. **e** Binding of ARNT with AHR in IL-6-pretreated THLE-2 cells with incubation of Leflunomide (LEF) and vehicle (DMSO) in the presence of *CRP* siRNA or negative control siRNA (NC siRNA), respectively. Pretreatment of IL-6 induced the overexpression of CRP in THLE-2 cells. **f** Relative levels of *CRP* mRNA (left) and CRP release (right) in IL-6-pretreated THLE-2 cells incubated with vehicle (DMSO), Leflunomide (LEF) and A77 1726, respectively. *$P < 0.05$ as determined by one-way ANOVA with a post hoc test. NS: no significance. Each experiment was repeated three times. **g** Besides the well-known immunomodulatory action via A77 1726, Leflunomide itself could activate AHR genomic signaling to inhibit CRP production and then attenuate bone erosion in CRP$^L$ RA (left), whereas the Leflunomide-AHR-CRP signaling was dysfunctional in CRP$^H$ RA (right). Source data are provided as a Source Data file

responsive target stability (DARTS) assay to confirm the binding between AHR and Leflunomide. DARTS assay relies on the protection against proteolysis of the target proteins conferred by the interaction with small molecular drugs[27]. Flag-AHR, Flag-AHR mutants and BSA were incubated with Leflunomide or DMSO (vehicle), followed by digestion with proteases (subtilisin), respectively (Supplementary Fig. 6a). Mutation of either H291 or K303 rather than V381 significantly decreased the stability of AHR conferred by the interaction with Leflunomide, suggesting that both H291 and K303 participated in the interaction between Leflunomide and AHR. Mutation of both H291 and K303 lead to synergetic reduction of AHR stability conferred by the interaction with Leflunomide (Supplementary Fig. 6b).

**Leflunomide-AHR-CRP signaling in hepatocytes.** Human normal hepatocyte cell line (THLE-2) was incubated with DMSO (vehicle), Leflunomide and A77 1726, respectively.

Immunoprecipitation analysis showed that Leflunomide, rather than A77 1726, induced binding of AHR with ARNT in a dose-dependent manner (Fig. 3c). Level of CRP expression and release were significantly decreased by Leflunomide but not by A77 1726 (Fig. 3d). After AHR knockdown, Leflunomide could not decrease CRP expression and release (Supplementary Fig. 6c, d). Leflunomide is mainly metabolized to A77 1726 by a cytochrome P450 1A2 enzyme (CYP1A2)[28]. Furafylline, a specific inhibitor of CYP1A2, could suppress metabolism of Leflunomide[28]. We incubated human normal primary hepatocytes with Leflunomide in the presence of Furafylline or not. Leflunomide induced binding of AHR with ARNT and decreased CRP expression and release. Stabilization of Leflunomide via inhibition of its metabolism by Furafylline further enhanced AHR-ARNT interaction and CRP inhibition (Supplementary Fig. 6e, f). All the above results demonstrated that Leflunomide itself but not A77 1726 induced AHR genomic signaling to inhibit CRP expression in normal hepatocytes in vitro. In addition to the genomic signaling,

AHR also functions through nongenomic pathways, which is independent of AHR-ARNT interaction[29]. Some ligands have been reported to trigger AHR nongenomic pathways to regulate activities of certain proteins, such as Src, JunD, and cyclin A[29,30]. Leflunomide had no effects on expression of JunD and cyclin A and Src activation (phosphorylation of Src) in THLE-2 cells (Supplementary Fig. 6g), indicating that Leflunomide specifically induced AHR genomic signaling in normal hepatocytes. To examine whether Leflunomide induced AHR genomic signaling to inhibit CRP in hepatocytes overexpressing CRP, THLE-2 cells were pretreated with IL-6 to induce overexpression of CRP[22]. Leflunomide showed weak ability in induction of AHR-ARNT interaction in THLE-2 cells overexpressing CRP. Knockdown of CRP recovered the ability of Leflunomide in induction of AHR-ARNT interaction (Fig. 3e). Leflunomide could not inhibit CRP expression and release in THLE-2 cells overexpressing CRP (Fig. 3f).

**Inhibition of CRP and bone erosion by Leflunomide itself**. Both $CRP^L$ and $CRP^H$ CIA rats were orally administered with vehicle, Leflunomide and A77 1726, respectively (Supplementary Fig. 7a). Levels of serum CRP and TRAP5b were lower in $CRP^L$ rats treated with Leflunomide than those treated with A77 1726 (Supplementary Fig. 7b). Inhibition of bone resorption and bone loss by Leflunomide was more significant in $CRP^L$ rats when compared to A77 1726 (Supplementary Fig. 7c, d). In contrast, Leflunomide could not significantly decrease CRP, bone resorption and bone loss in $CRP^H$ rats when compared to A77 1726 (Supplementary Fig. 7e-g). These results, together with the above data, leading us to postulate that besides the well-known immunomodulatory action of Leflunomide via A77 1726, Leflunomide itself could activate AHR genomic signaling to inhibit CRP production and then attenuate bone erosion in $CRP^L$ RA, whereas the Leflunomide-AHR-CRP signaling was dysfunctional in $CRP^H$ RA (Fig. 3g).

**HIF1α-induced dysfunction of Leflunomide-AHR-CRP signaling**. HIF1α has been reported to compete with AHR for ARNT association[31]. We examined the level of HIF1α in hepatocytes from non-immunized, $CRP^L$ and $CRP^H$ CIA rats. $CRP^L$ rats had slightly elevated HIF1α expression when compared to non-immunized rats, while HIF1α level in $CRP^H$ rats was significantly increased (Fig. 4a). There was a positive association between CRP and *HIF1α* expression in both $CRP^L$ and $CRP^H$ rats (Supplementary Fig. 8a). In addition, CRP inhibition by LNPs-*CRP* siRNA led to a reduction of HIF1α expression in hepatocytes from $CRP^H$ rats (Supplementary Fig. 8b). We manipulated CRP by siRNA or overexpressing vectors in THLE-2 cells in vitro. Knockdown of CRP decreased HIF1α expression and binding of ARNT with HIF1α (Fig. 4b, c, Supplementary Fig. 8c). THLE-2 cells with CRP overexpression showed increase of HIF1α and binding of ARNT with HIF1α (Fig. 4d, e, Supplementary Fig. 8d). We examined AHR genomic signaling stimulated by Leflunomide in THLE-2 cells transfected with CRP siRNA or overexpressing vectors. Knockdown of CRP enhanced AHR-ARNT interaction, whereas CRP overexpression reduced the interaction between ARNT and AHR in THLE-2 cells upon Leflunomide stimulation (Fig. 4f, g). Binding of AHR with ARNT and inhibition of CRP expression induced by Leflunomide was remarkable in hepatocytes from $CRP^L$ CIA rats but insignificant in hepatocytes from $CRP^H$ CIA rats (Supplementary Fig. 8e, f). We concluded that high expression of CRP could trigger upregulation of HIF1α, which competed with AHR for ARNT association, leading to the dysfunction of Leflunomide-AHR-CRP signaling in $CRP^H$ RA (Fig. 4h). We determined whether knockdown of HIF1α could

improve Leflunomide-AHR-CRP signaling in THLE-2 cells overexpressing CRP in vitro. Knockdown of HIF1α decreased the binding of ARNT with HIF1α (Fig. 4i). *HIF1α* siRNA alone had not effects on AHR-ARNT interaction and CRP level(Fig. 4j, k). However, *HIF1α* siRNA in combination with Leflunomide effectively enhanced AHR genomic signaling and CRP inhibition (Fig. 4j, k), suggesting that inhibition of HIF1α could facilitate Leflunomide activating AHR to inhibit CRP expression in hepatocytes in $CRP^H$ RA.

**Limited efficacy of Leflunomide in $CRP^H$ CIA mice**. DBA/1 mice with CIA were administered with Leflunomide for 28 days (Supplementary Fig. 9a). The mice were also classified into PBE+ and PBE− subgroups based on microCT analysis. The deteriorated bone microarchitecture was observed in PBE+ CIA mice rather than in PBE− CIA mice after Leflunomide treatment (Supplementary Fig. 9b). Inhibition of DHODH activity and proliferation of immune cells (T and B lymphocytes and macrophages), as well as decrease of cytokines (IL-17, IL-6, and RANKL) were comparable between PBE− and PBE+ mice (Supplementary Fig. 9c). The PBE + mice demonstrated higher serum baseline CRP ($CRP^H$, $34.67 \pm 5.41$ mg L$^{-1}$) and TRAP5b, whereas the PBE- mice showed relatively lower CRP ($CRP^L$, $17.70 \pm 3.25$ mg L$^{-1}$) and TRAP5b (Supplementary Fig. 9d). Leflunomide decreased levels of serum CRP and TRAP5b and inhibited bone resorption in $CRP^L$ mice but not in $CRP^H$ mice (Supplementary Fig. 9e, f).

**Re-activation of Leflunomide by hepatic *HIF1α* deletion**. $HIF1α^{loxP/loxP}$ mice were bred to *Alb-cre* mice to generate hepatocyte-specific *HIF1α* knockout (*HIF1α*-HKO) mice (Supplementary Fig. 10a, b). Serum CRP level was comparable between *HIF1α*-HKO and control ($HIF1α^{loxP/loxP}$) mice (Supplementary Fig. 10c), suggesting that HIF1α could not affect CRP expression in hepatocytes in vivo. CIA was established in both *HIF1α*-HKO and control mice. They were classified into subgroups based on the above established reference range of CRP for $CRP^L$ ($17.70 \pm 3.25$ mg L$^{-1}$) and $CRP^H$ CIA mice ($34.67 \pm 5.41$ mg L$^{-1}$). After Leflunomide treatment, we observed the attenuated bone erosion in $CRP^L$ but not in $CRP^H$ control mice. However, both $CRP^L$ and $CRP^H$ *HIF1α*-HKO mice demonstrated satisfactory response to Leflunomide in attenuating bone erosion (Supplementary Fig. 10d, e). Inhibition of inflammatory synovial hyperplasia by Leflunomide in $CRP^L$ and $CRP^H$ subgroups was comparable for both control and *HIF1α*-HKO CIA rats (Supplementary Fig. 10f). Leflunomide decreased CRP and TRAP5b and inhibited osteoclastic activity and bone resorption in $CRP^L$ but not in $CRP^H$ control mice, while decrease of CRP and TRAP5b and inhibition of osteoclastic activity and bone resorption by Leflunomide were significant in both $CRP^L$ and $CRP^H$ *HIF1α*-HKO CIA rats (Supplementary Fig. 10g-i). All the above data suggested that HIF1α in hepatocyte was a molecular target for re-activating Leflunomide inhibiting CRP and bone erosion in $CRP^H$ RA.

**Re-activation of Leflunomide-AHR-CRP signaling by ACF**. THLE-2 cells overexpressing CRP were treated with DMSO (vehicle) or a HIF1α inhibitor ACF[32]. ACF reduced binding of ARNT with HIF1α and downregulated the most classical target gene of HIF1α-ARNT signaling, i.e., vascular endothelial growth factor (*VEGF*)[33] (Fig. 5a). THLE-2 cells overexpressing CRP were incubated with DMSO (vehicle), Leflunomide, ACF and combination of Leflunomide and ACF, respectively. The combination of Leflunomide and ACF enhanced binding of ARNT with AHR and reduced CRP expression and release when compared to Leflunomide or ACF

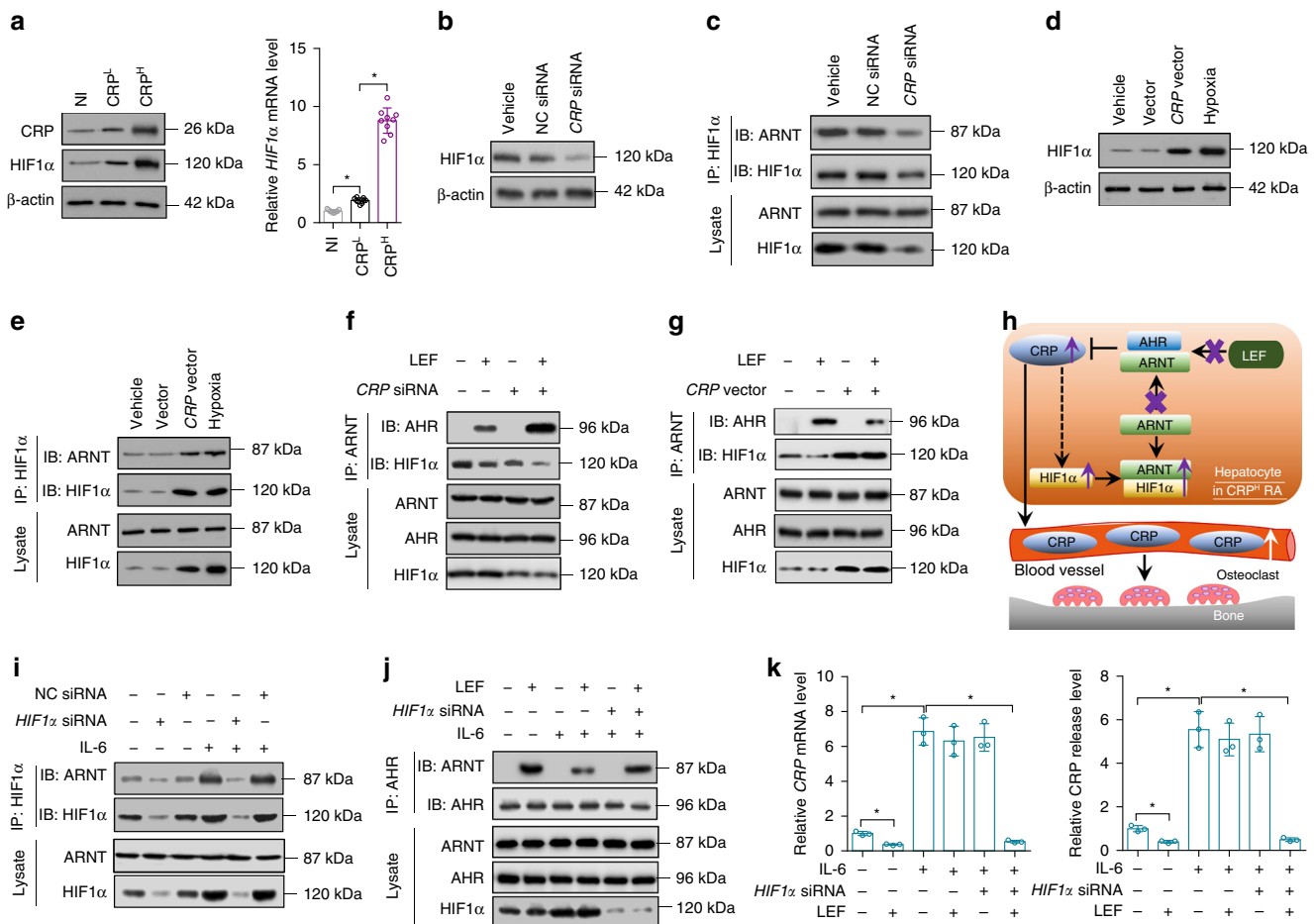

**Fig. 4** CRP upregulated HIF1α to interfered Leflunomide-AHR-CRP signaling in vitro. **a** Levels of CRP and HIF1α in hepatocytes from non-immunized (NI) rats, CRP[H] and CRP[L] CIA rats. $n = 9$ for each group. **b** Level of HIF1α in human normal hepatocytes (THLE-2 cells) transfected with vehicle, *CRP* siRNA and negative control siRNA (NC siRNA), respectively. **c** Level of ARNT binding with HIF1α in THLE-2 cells after the transfection. **d** Level of HIF1α in THLE-2 cells transfected with vehicle, empty vector and *CRP* overexpressing vector, respectively. THLE-2 cells incubated in hypoxia were used as a positive control. **e** Level of ARNT binding with HIF1α in THLE-2 cells after the transfection. **f** Level of ARNT binding with HIF1α or AHR in THLE-2 cells transfected with *CRP* siRNA in the presence of Leflunomide (LEF). **g** Level of ARNT binding with HIF1α or AHR in THLE-2 cells transfected with *CRP* overexpressing vector in the presence of Leflunomide (LEF). **h** Proposed mechanism underlying the dysfunction of Leflunomide-AHR-CRP signaling in CRP[H] RA. Briefly, high level of CRP could trigger upregulation of HIF1α, which competes with AHR for ARNT association, leading to the dysfunction of Leflunomide-AHR-CRP signaling in CRP[H] RA. **i** Level of ARNT binding with HIF1α in IL-6-pretreated THLE-2 cells transfected with *HIF1α* siRNA and NC siRNA, respectively. Pretreatment of IL-6 induced the overexpression of CRP in THLE-2 cells. **j** Level of ARNT binding with AHR in IL-6-pretreated THLE-2 cells transfected with *HIF1α* siRNA in the presence of Leflunomide (LEF). **k** Relative *CRP* mRNA (left) and CRP release (right) in IL-6-pretreated THLE-2 cells after the treatment. *$P < 0.05$ as determined by one-way ANOVA with a post hoc test. Each experiment was repeated three times. Source data are provided as a Source Data file

alone (Fig. 5b, c). We collected the conditioned mediums from the above treatment groups and incubated the monocytes with the conditioned mediums in the presence of macrophage colony stimulating factor (M-CSF) (Fig. 5d). Osteoclast differentiation were inhibited in monocytes treated with conditioned medium from the combination of Leflunomide and ACF, but not in those cells treated with conditioned medium from Leflunomide or ACF alone group (Fig. 5e–g). These results demonstrated that ACF facilitated Leflunomide activating AHR to suppress CRP expression and decrease osteoclast differentiation in vitro.

**Decrease of bone erosion by Leflunomide combined with ACF.** To investigate whether ACF facilitated Leflunomide decreasing CRP production and attenuating bone erosion in CRP[H] CIA rats, CRP[H] rats were administered with vehicle, Leflunomide, ACF and the combination of Leflunomide and ACF, respectively (Fig. 6a). CRP[H] rats administered with the combination of Leflunomide and ACF showed reduced serum CRP, attenuated

bone erosion and bone resorption and prevented bone loss, whereas no obvious decrease of CRP and bone erosion were observed in CRP[H] rats administered with Leflunomide or ACF alone (Fig. 6b–e and Supplementary Fig. 11). Leflunomide or the combination of Leflunomide and ACF attenuated inflammatory synovial hyperplasia in CRP[H] CIA rats (Supplementary Fig. 11). There was no significant change in liver function parameters including alanine aminotransferase (ALT), asporate aminotransferase (AST), a kidney function parameter blood urea nitrogen (BUN) and hematology parameters including total protein (TP), hemoglobin, white blood cell (WBC), red blood cell (RBC), and platelet (PLT) in CRP[H] CIA rats treated with the combination of Leflunomide and ACF when compared to Leflunomide or ACF alone (Supplementary Table 5).

## Discussion

In our study, we found that Leflunomide exerts significant immunosuppression but limited efficacy in attenuating bone erosion in

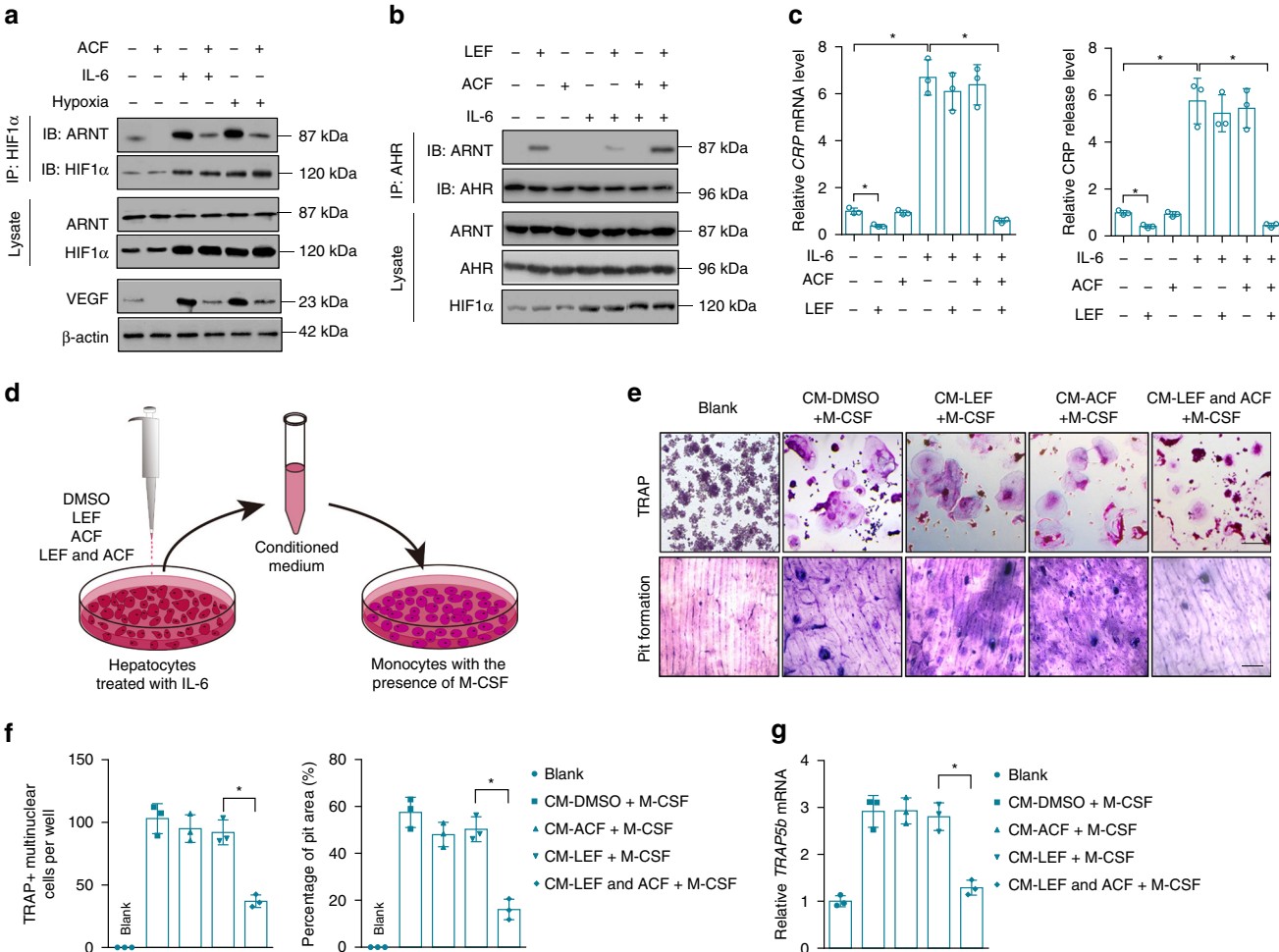

**Fig. 5** ACF re-activated Leflunomide-AHR-CRP signaling in vitro. **a** Binding of ARNT with HIF1α and level of VEGF in IL-6-pretreated THLE-2 cells incubated with vehicle (DMSO) or ACF (5.0 μM). Pretreatment of IL-6 induced the overexpression of CRP in THLE-2 cells. THLE-2 cells incubated in hypoxia were used as a positive control. **b** Binding of ARNT with AHR in IL-6-pretreated THLE-2 cells incubated with vehicle (DMSO), Leflunomide (LEF, 10 μM), ACF (5.0 μM) and the combination of Leflunomide and ACF (LEF + ACF, LEF, 10 μM; ACF, 5.0 μM), respectively. **c** Relative *CRP* mRNA and CRP release in IL-6-pretreated THLE-2 cells after the treatment. **d** The diagram of the experimental design. Briefly, THLE-2 cells were pretreated with IL-6 and then incubated with vehicle (DMSO), Leflunomide (LEF, 10 μM), ACF (5.0 μM) and the combination of Leflunomide and ACF (LEF + ACF, LEF, 10 μM; ACF, 5.0 μM), respectively. After 8 h, the medium was replaced, and the THLE-2 cells were continuously cultured for another 48 h. The conditioned mediums were collected for culturing monocytes in the presence of macrophage colony stimulating factor (M-CSF). **e** TRAP staining (upper) and pit formation assay (bottom) in monocytes incubated with the conditioned mediums. Scale bar, 100 μm. **f** Quantitative determination of TRAP-positive (TRAP+) multinuclear cells per well and percentage of pit area from pit formation assay. **g** Relative *TRAP5b* mRNA level in monocytes incubated with the conditioned mediums. *$P < 0.05$ as determined by one-way ANOVA with a post hoc test. Each experiment was repeated three times. Source data are provided as a Source Data file

RA individuals (PBE+) distinguished by higher serum CRP (CRP^H). whereas the others (PBE−) with satisfactory responsiveness to Leflunomide show relatively lower serum CRP (CRP^L).

The exact role of CRP in RA has been an unsettled dispute for a long time. No human study to date has directly investigated the contribution of CRP in RA, and animal studies performed so far have shown mixed results. CRP has been reported to induce prolonged inflammation in arthritic rabbits and directly promote osteoclastogenesis[10,34], indicating that CRP makes a detrimental contribution to RA. However, transgenic expression of human and rabbit CRP has been shown to inhibit CIA development[35,36], while knockout of CRP exacerbates CIA and bone damage[11,35], suggesting that CRP plays a protective role in RA. In our study, we demonstrate the detrimental function of CRP in CIA rats by inhibition of CRP with anti-CRP antibodies or liver-targeted LNPs-*CRP* siRNA. We compare the above studies which seem to draw incompatible conclusions about the role of CRP in RA.

Regarding the studies demonstrating the protective role of CRP in RA[11,35,36], CRP expression is manipulated prior to arthritic induction in genetic models (CRP transgenic and knockout mice). For our work and other studies demonstrating the detrimental role of CRP in RA[34], however, CRP level is influenced after arthritic induction in CIA models. The above studies actually reveal the different roles of CRP in distinct stages of RA development (early induction stage versus late active stage). Together, they suggest that CRP may have double-faced functions in RA. CRP may confer benefit during the early induction phase of RA but play a detrimental role during active RA, which is consistent with the assumption in one of the above study[35].

Previously, Kim et al. report that CRP promotes osteoclastic activities via FcγRs signaling with the absence of RANKL[10], while Cho et al. show that CRP inhibits RANKL-induced osteoclastic activities in vitro[37]. Recently, Jia et al. suggest that CRP regulates osteoclastic differentiation in conformation-

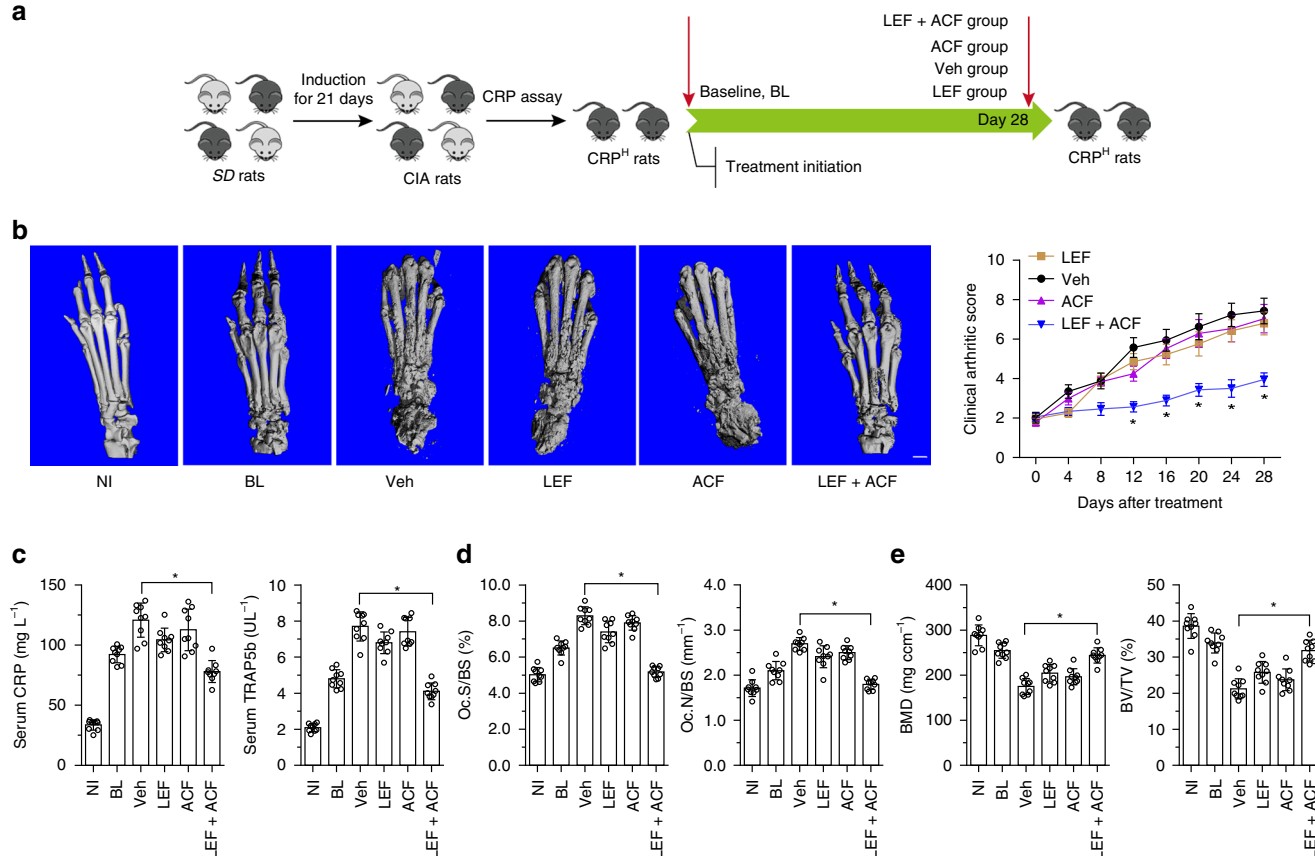

**Fig. 6** ACF facilitated Leflunomide attenuating bone erosion in CRP[H] CIA rats. **a** The diagram of the experimental design. Briefly, *SD* rats were immunized with bovine type II collagen for 21 days to establish the CIA model. The CIA rats with CRP level within the established reference range for CRP[H] rats (91.24 ± 10.12 mg L[−1]) were administered with vehicle (Veh), Leflunomide (LEF, 10.0 mg kg[−1] per day), ACF (1.0 mg kg[−1] per day), and the combination of Leflunomide and ACF (LEF + ACF, LEF, 10,0 mg kg[−1] per day; ACF, 1.0 mg kg[−1] per day) for 28 days, respectively. **b** The representative reconstructed three-dimensional micro-CT images and clinical arthritic score of the hind paws from CRP[H] rats before (baseline, BL) and after the treatment. Non-immunized (NI) rats were used as the controls of the CIA rats. Scale bar, 5.0 mm. *$P < 0.05$ as determined by repeated - measures ANOVA with a post hoc test. **c** Levels of serum CRP and TRAP5b in CRP[H] rats before (baseline, BL) and after the treatments. **d** Bone resorption parameters including osteoclast surface per bone surface (Oc.S/BS) and osteoclast number per bone surface (Oc.N/BS) in CRP[H] rats before (baseline, BL) and after the treatments. **e** Bone mass parameters including bone mineral density (BMD) and bone volume per total volume (BV/TV) in CRP[H] rats before (baseline, BL) and after the treatments. *$P < 0.05$ as determined by one-way ANOVA with a post hoc test. $n = 9$ for each group. Source data are provided as a Source Data file

and RANKL-dependent manners[11]. Circulating native CRP is composed of five identical subunits but dissociates into the monomeric conformation upon entering local lesions[21]. Jia et al. show that monomeric CRP could induce osteoclastic differentiation with the absence of RANKL in vitro[11], which is consistent with Kim's report[10]. However, Jia et al. also demonstrate that monomeric CRP inhibits RANKL-induced osteoclastic differentiation by neutralizing RANKL in vitro[11], which is consistent and accounts for Cho's findings[37]. Jia et al. further explain that native CRP is inefficient for osteoclastic activity in vitro but the prolonged incubation time (3–21 days) in cell culture would favor the conversion of native CRP to monomeric CRP[11], suggesting that in both Kim and Cho's studies, CRP regulates osteoclastic activity actually via the monomeric conformation, as they incubate cells with CRP for 3–21 days and 3–5 days, respectively[10,37]. Based on these studies, we think whether CRP promotes or inhibits osteoclastic activities in vivo may be dependent on ratio of monomeric CRP to RANKL. We find that molar concentration of the monomeric CRP is over 10,000-fold of RANKL in both CRP[L] and CRP[H] RA patients, indicating that the monomeric CRP is enough to neutralizing RANKL and the redundant free monomeric CRP would dominate osteoclastic activities in RA.

Previously, pertinent publications frequently reiterate that mouse CRP is a trace protein and serum levels of CRP in mice are markedly lower than in humans and rabbits, <3 mg mL[−1] even after inflammatory stimuli[36,38]. However, recent studies argue that such statements are not convincing as they either refer to review articles devoid of experimental data or an early publication, which fails to detect any mouse CRP using sheep anti-human CRP antibodies cross-reacting with mouse CRP before acute phase induction[39,40]. To date, reliable reagents specific for mouse have become available and studies suggest that serum CRP levels may be vastly underestimated[40,41]. High serum CRP levels ranging from 5 mg mL[−1] to 50 mg mL[−1] have been determined in different mouse models[40,42–46], which are consistent with our results.

In our work, *Sprague-Dawley* (*SD*) rats and *DBA/1* mice are used for establishment of CIA, as they are common animal models for exploring the pathogenic mechanism of RA and therapeutic action of anti-arthritic agents[47]. *SD* rat is an outbred strain and *DBA/1* mouse is an inbred strain[48,49]. We find the large variations of serum CRP levels in both *SD* rats and *DBA/1* mice, which are the basis for classification of CIA models into CRP[L] and CRP[H] subgroups. For many decades, inbred rats and mice are preferred over outbred strains because it is assumed that

they display lower levels of variations due to their greater homogeneity[50]. However, very few studies with adequate sample sizes have in fact compared phenotypic variations between inbred strains and outbred strains[51]. Surprisingly, recent reports suggest that inbred and outbred animals show comparable phenotypic variations[50–52]. On one hand, they conclude that outbred animals could be widely adopted as research subjects like inbred animals. On the other hand, it could not be ignored that the coefficients of variation vary in large ranges for both of them[50]. We review other studies, which also show large variations of serum CRP[42–44]. In our previous work, we also demonstrate the large variations of other proteins in signaling pathways[53]. Even though the large variations of proteins in established animal models seem to differ from the common sense and the underlying reasons are still unknown, the variations could be used for explaining non-response of therapeutics in human disease[53].

## Methods

**Characteristics of RA patients treated with Leflunomide**. 250 RA patients (150 women and 100 men) were treated with Leflunomide. The characteristics of the patients before treatment were shown in Supplementary Table 1. The patients with other diseases or medication, including calcium supplements and postmenopausal hormone replacement therapy, that might affect bone remodeling, were excluded from the study. Female patients who were postmenopausal for fewer than 5 years were also excluded from the study. Informed consent was obtained from all patients. The study was approved by the Ethics Committee in Institute of Basic Research in Clinical medicine, China Academy of Chinese Medical Sciences (CACMS). All the included RA patients were orally administrated with Leflunomide for 12 months. The serum from all the RA patients was collected before and after treatment. The anteroposterior plain radiographs of hands were taken before and after treatment.

**Radiological evaluation by modified Sharp's method**. The anteroposterior plain radiographs of the hands were read and evaluated by two different qualified doctors. The reader of the radiographs was blind to the order of the radiographs and the clinical details of the RA patients. The erosion score per joint of the hands was performed according to the modified Sharp's score system. Briefly, the erosions for 32 joints of both hands were evaluated based on a 6-points scale (scale 0–5): Erosions were scored 0 if there was no erosion. Erosions were scored 1 if they were discrete but clearly present and 2 or 3 if they were larger, depending on the surface area of the joint involved. A score of 3 was given if the erosion was large and extends over the imaginary middle of the bone. A score of 5 was given if a complete collapse of the joint was present or if the full surface of the joint is affected[54].

**Cell culture**. Human normal hepatocyte cell line THLE-2 (ATCC, CRL-2706) was cultured at 37 °C in a humidified environment containing 5% $CO_2$ and 20% $O_2$ in bronchial epithelial cell basal medium and additives (BEGM from Lonza/Clonetics Corporation, USA, BEGM Bullet Kit). Medium was changed every two days after enzymatic dissociation and centrifugation at $300 \times g$ for 5 min. No mycoplasma contamination was confirmed for the cell line. There was no commonly mis-identified cell line used in this study. THLE-2 cells ($2 \times 10^5$ cells per well) were incubated overnight. For knockdown of CRP or AHR, the cells were transfected with specific siRNAs or NC siRNA using X-tremeGENE™ siRNA Transfection Reagent (Roche Life Science). The siRNAs against AHR and HIF1α and NC siRNA were from Thermo Fisher Scientific. The siRNA against CRP and NC siRNA were from Santa Cruz Biotechnology. For vector-induced overexpression of CRP, human CRP cDNA was PCR amplified using the primers: 5′-TGAATTCAGG CCCTTGTATC-3′ (sense) and 5′-TCCCAGCATAGTTAACGAGC-3′ (antisense). The complete nucleotide sequence of CRP was cloned into the mammalian expression vector pcDNA3.1 (Invitrogen) and confirmed by DNA sequencing. THLE-2 cells were transfected with pcDNA3.1 vector overexpressing CRP or empty vectors using X-tremeGENE™ HP DNA Transfection Reagent (Roche Life Science). After the transfection, the cells were continuously cultured for another 36 h. Small molecules at the indicated concentrations was added in the cells for 24 h. For IL-6-stimulated overexpression of CRP, the cells were pretreated with human recombinant IL-6 (50.0 ng mL$^{-1}$, R&D Systems) for 8 h. For hypoxic treatment, cells were exposed to 5% $CO_2$ and 1% $O_2$ for 16 h. Leflunomide, A77 1726, Furafylline and ACF were from Sigma-Aldrich.

**Expression and purification of Flag-tagged proteins**. Site mutations in AHR (H291A, K303A, V381A, and H291A/K303A) were made by the QuikChange site-directed mutagenesis kit (Agilent Technologies). AHR or the mutants was cloned into Flag-tagged mammalian expression vector pcDNA3.1 (Addgene). The vectors were confirmed by direct DNA sequencing and transfected into THLE-2 cells using X-tremeGENE™ HP DNA Transfection Reagent (Roche Life Science). After

transfection, the cells were continuously cultured for another 36 h. Flag-AHR and the mutants were purified by a FLAG® M purification kit (Sigma-Aldrich).

**Immunoprecipitation and western blots**. Cell lysates were prepared in HEPES lysis buffer (20 mM HEPES pH 7.2, 50 mM NaCl, 0.5% Triton X-100, 1 mM NaF, 1 mM dithiothreitol) supplemented with protease inhibitor cocktail (Roche Life Science) and phosphatase inhibitors (10 mM NaF and 1 mM Na3VO4). Immunoprecipitation was performed using primary antibodies and protein A/G-agarose (Santa Cruz Biotechnology) at 4 °C. For western blots, protein samples were separated by sodium dodecyl sulfate-polyacrylamide gel electrophoresis and transferred onto PVDF membranes (Bio-Rad). After blocking, the membranes were probed with primary antibodies and then incubated with specific horseradish peroxidase-conjugated secondary antibodies (Bio-Rad). Immunodetection was performed using enhanced chemiluminescence (Thermo Fisher Scientific). β-actin was used as a loading control for internal correction. The primary antibodies included an anti-AHR antibody (ab2769, 1: 500), an anti-ARNT antibody (ab5624, 1: 500), an anti-HIF1α (ab1, 1: 200) antibody, an anti-CRP antibody (ab50861, 1: 2000), an anti-VEGFA antibody (ab46154, 1: 1000), an anti-JunD antibody (ab28837, 1: 500), an anti-cyclin A antibody (ab181591, 1: 2000), an anti-Src antibody (ab47405, 1: 500), an anti-Src (phospho Y418) antibody (ab4816, 1: 1000) and an anti-β-actin (ab8226, 1: 10000) antibody from Abcam. Antibodies were validated for the specific species and application in pilot studies. All relevant uncropped blots are available in the Source Data file.

**Isolation of lymphocytes and macrophages**. The synovial fluid was obtained from RA patients, CIA rats and mice. T lymphocytes were isolated by EasySep™ T Cell Isolation Kits (STEMCELL Technologies Inc.). B lymphocytes were isolated by EasySep™ B Cell Isolation Kits (STEMCELL Technologies Inc.). The purity of T and B lymphocytes as determined by flow cytometry was approximately 90–95%. Macrophages were isolated by flow cytometry using an anti-CD14 antibody (17000-1-AP, 1: 100, Proteintech Group, Inc).

**DHODH activity assay**. The assay comprised enzymatic reaction of DHODH with dihydroorotic acid (DHO) substrate, followed by fluorescence detection specific for orotic acid using the 4-trifluoromethyl-benzamidoxime (4-TFMBAO) fluorogenic reagent[55]. Briefly, $3 \times 10^5$ lymphocytes were lysed by sonication for 10 min at 4 °C. Lysates were incubated with 1.0 ml aqueous solution containing 500 μM DHO, 200 mM $K_2CO_3$-HCl (pH 8.0), 0.2% Triton X-100 and 100 μM coenzyme Q10 at 37 °C for 1 h. Hundred microliter aliquot was mixed with 150 μL $H_2O$, 250 μL of 4.0 mM 4-TFMBAO, 250 μL of 8.0 mM $K_3[Fe(CN)_6]$ and 250 μL of 40 mM $K_2CO_3$ (pH 11) and then heated at 80 °C for 4.0 min. The reaction was stopped by cooling in an ice-water bath and the intensity was measured with a spectrofluorometer. Excitation and emission wavelengths were 340 nm and 460 nm, respectively. For the calculation of the production, the preexisting level of orotic acid at non-incubation time was subtracted from total amount of orotic acid observed at the end of the hour incubation. Orotic acid, coenzyme Q10, 4-TFMBAO and DHO were purchased from Sigma-Aldrich.

**Isolation and osteoclastogenesis of human monocytes**. Peripheral blood mononuclear cells (PBMCs) were isolated from heparinized whole blood of healthy volunteers by density gradient centrifugation using Ficoll-Hypaque (Sigma-Aldrich). The cells were washed three times with sterile PBS and resuspended in RPMI 1640 (Life Technologies) supplemented with 10% FBS, 2 mM L-glutamine, and 1% penicillin-streptomycin (complete medium). The PBMCs were incubated at 37 °C in complete medium and allowed to adhere for 45 min. The nonadherent cells were removed and the adherent cells were washed with sterile PBS and harvested with a rubber policeman. The purity of the monocytes isolated by Ficoll-Hypaque as determined by flow cytometry was approximately 85–90%. The cells were cultured for three weeks in conditioned mediums from THLE-2 cells in the presence of 25 ng mL$^{-1}$ recombinant human M-CSF (R&D Systems). The medium was changed every other day. On day 21, TRAP-positive cells were identified using a leukocyte acid phosphatase kit (Sigma-Aldrich)[10].

**Isolation of hepatocytes**. Rat and mouse primary hepatocytes were isolated using a two-step perfusion process[56]. Briefly, hepatocytes are dissociated from anesthetized adult rats by a non-recirculating collagenase perfusion through the portal vein. The isolated cells are then filtered through a 100 μm pore size mesh nylon filter. Cell number within the cell suspension was counted by using a hemocytometer. Cell viability was determined by trypan blue staining. Flow cytometry was used for purification and identification of the isolated hepatocytes.

**Pit formation assay**. Human monocytes were seeded on OsteoAssay bone plates (Lonza) at a density of $1 \times 10^4$ cells per well. The cells were cultured for three weeks in conditioned mediums from THLE-2 cells in the presence of 25 ng mL$^{-1}$ recombinant human M-CSF (R&D Systems). The medium was changed every other day. On day 21, bone slices were ultra-sonicated in 1 M $NH_4OH$ to remove adherent cells and stained with 0.1% toluidine blue solution. Bone slice images were captured using electron microscopy. Three fields were randomly selected for each

bone slice for further analysis. Pit areas were quantified using Image Pro Plus 6.2 software (Media Cybernetics Inc.)[57].

**Molecular docking**. Position-Specific Iterated BLAST (PSI-BLAST) against PDB database was performed to identify homology templates of AHR PAS-A and PASB domains. Homology modeling were performed by MODELER software to build 3D structure of AHR PAS-A and PAS-B domains. Structure with lowest energy was chosen as the starting point for further structure refinements. In order to find optimized conformations, loop refinement was applied to the predicted structure of AHR PAS-A and PAS-B domains[58]. Ramachandran plot was used to estimate the quality of modeled structure by inspecting backbone dihedral angles ψ against φ of amino acid residues in protein structure. Molecular docking was conducted between AHR PAS-A and PAS-B domains and Leflunomide using Autodock Vina. Once the macromolecule structure of AHR is loaded, charges and hydrogen atoms were added, and non-polar hydrogen atoms were merged. Leflunomide molecule was drawn using ChemDraw 10.0 and structure were optimized using quantum chemistry method AM1. Both protein and ligand structures were converted to PDBQT format using AutoDock tools.

**ELISA assay**. The human and rat CRP were measured by ELISA kits from Thermo Fisher Scientific. The mouse CRP was measured by a Quantikine ELISA kit (R&D Systems). The human rheumatoid factors including IGA, IGM and IGG were determined by ELISA kits from Cusabio Biotech. The human IL-6 and RANKL were determined by a PeliKine[TM] human ELISA kit from Cell Sciences and an ELISA kit from Boster Biological Technology, respectively. The human, rat and mouse IL-17 was determined by ELISA kits from Abcam. The human TRAP5b was determined by an ELISA kit from TECOmedical AG. The mouse and rat TRAP5b were determined by ELISA kits from Immunodiagnostic Systems, respectively. The mouse and rat IL-6 was determined by ELISA kits from Abcam. The mouse and rat RANKL were determined by ELISA kits from Abcam and Biobyt. The monomeric CRP was determined by an ELISA assay[59].

**Serum biochemistry and hematology assays**. The rat liver function parameters (ALT and AST) and a kidney function parameter (BUN) were analyzed by a Vitros 250 Analyzer (Ortho Clinical Diagnostics, Johnson & Johnson Co, Rochester, NY). The hematology parameters including TP, hemoglobin, WBC, RBC and PLT were analyzed by an ABX Pentra 60 C + Analyzer (Horiba ABX, Montpellier, France).

**Preparation of LNPs-siRNA**. LNPs delivery systems comprised lipid-like 98N$_{12}$-5 (1), cholesterol, PEG-lipid and siRNA[12,24]. Briefly, stock solutions of 98N$_{12}$-5(1), mPEG2000-lipid, and cholesterol (Sigma-Aldrich) were prepared in ethanol and mixed to yield the molar ratios at 42:48:10. Mixed lipids were added to 125 mM sodium acetate buffer (pH 5.2) to yield a solution containing 35% ethanol, resulting in spontaneous formation of empty lipidoid nanoparticles. The resultant nanoparticles were extruded through a 0.08 μm membrane (Sterlitech, Kent, WA) using a LIPEX Extruder (Northern Lipids, Burnaby, British Columbia, Canada) to form particles 50–60 nm in length. CRP siRNA or NC siRNA in 50 mM sodium acetate (pH 5.2) and 35% ethanol was added to the nanoparticles at the total lipid:siRNA ratios at 7.5:1(wt:wt) and incubated at 37 °C for 30 min. Ethanol removal and buffer exchange of siRNA-containing nanoparticles was achieved by dialysis against phosphate-buffered saline. Finally, the formulation was filtered through a 0.2 μm sterile filter. For determining tissue distribution of the siRNA, siRNA was labelled with Cy5 and fluorescence intensity in different organ samples was measured by a microplate reader system (Bioscan, Washington, DC).

**CIA rat model**. Male *SD* rats (170–190 g of weight) were from the Chinese University of Hong Kong and kept with free access to food and water under standard temperature conditions (22 °C) and a 12 h light/dark cycle. CIA rat model was induced[60]. Briefly, bovine type II collagen (C II) (Chondrex) was emulsified in an equal volume of Incomplete Freund's adjuvant (IFA, Chondrex). The rats were immunized subcutaneously at the base of the tail with 200 μL emulsion (200 μg of bovine C II). Booster immunization was administered on day 7 with 100 μL injection of same emulsion as the first time. Arthritis severity was evaluated by clinical arthritic scores which were performed by two independent, blinded observers. Scoring was performed with a 0–4 scale, where 0 = no swelling or erythema; 1 = slight swelling and/or erythema; 2 = low-to-moderate edema; 3 = pronounced edema with limited joint usage; and 4 = excess edema with joint rigidity. Each hind limb was graded and thus the maximum possible score was 8 for each rat. A rat with a score of one or above was regarded as arthritic. All experimental procedures were approved by the Committees of Animal Ethics and Experimental Safety of the Hong Kong Baptist University. We complied with all relevant ethical regulations for animal testing and research.

**Generation of hepatocyte-specific *HIF1α* knockout mice**. *HIF1α*$^{loxP/loxP}$ mice (B6.129-*Hif1a*$^{tm3Rsjo}$/J; Jackson Laboratory) were from the Laboratory Animal Services Centre of the Chinese University of Hong Kong. *Alb-cre* mice (B6.Cg-Speer6-ps1$^{Tg(Alb-cre)21Mgn}$/J) were from the Jackson Laboratory. The above two mice were on a *C57BL/6* background, which is regarded to be relatively resistant to

arthritis induction[61]. Even though *C57BL/6* mice could develop arthritis after immunization with chicken-derived collagen type II, the maximum incidence of arthritis was relatively lower (50%-70%), and the arthritis susceptibility was variable[61]. Thus, the two mice were backcrossed for eight generations to change the genetic background from *C57BL/6* to *DBA/1*, which showed the greatest degree of susceptibility to arthritis induction[61–63]. Then, the *HIF1α*$^{loxP/loxP}$ mice were bred to *Alb-cre* mice to generate the hepatocyte-specific *HIF1α* knockout (*HIF1α*-HKO) mice[64].

**CIA mouse model**. Male *DBA/1* mouse aged 8–10 weeks were from the Chinese University of Hong Kong and kept with free access to food and water under standard temperature conditions (22 °C) and a 12 h light/dark cycle. CIA mouse model was induced[63]. Briefly, bovine C II (Chondrex) was emulsified in an equal volume of Complete Freund's adjuvant (CFA). The mice were immunized with single subcutaneous injection at the base of the tail with 100 μL emulsion (100 μg of bovine C II) containing a final concentration of 2 mg mL$^{-1}$ of *Mycobacterium tuberculosis*. Arthritis severity was evaluated by clinical arthritic scores which were performed by two independent, blinded observers. Scoring was performed with a 0–4 scale, where 0 = normal; 1 = redness and/or swelling in one joint; 2 = redness and/or swelling in more than one joint; 3 = redness and/or swelling in the entire paw; and 4 = deformity and/or ankylosis. Each hind limb was graded and thus the maximum possible score was 8 for each mouse. A mouse with a score of one or above was regarded as arthritic. All the experimental procedures were approved by the Committees of Animal Ethics and Experimental Safety of the Hong Kong Baptist University. We complied with all relevant ethical regulations for animal testing and research.

**Drug administration**. Leflunomide and A77 1726 were suspended in 0.5% carboxymethyl cellulose sodium (CMC-Na) and orally administrated to CIA rats or mice. ACF were dissolved in PBS and intraperitoneally administrated to the CIA rats. Rabbit IgG controls (ab171870, 20 μg for each rat, Abcam) or anti-CRP antibodies (ab227507, 20 μg for each rat, Abcam) were intra-articularly injected to the CIA rats.

**Histological analysis**. Mouse and rat knee joints were removed postmortem, stored in 10% neutral formalin, decalcified in 20% ethylenediaminetetraacetic acid (EDTA) for 4 weeks, then dehydrated and embedded in paraffin. The sections were stained with hematoxylin & eosin (HE) and tartrate-resistant acid phosphatase (TRAP) before histopathological analysis[65].

**MicroCT analysis**. For in vivo microCT analysis, the rats or mice were anesthetized and placed prone on a scan stage before scanning. A vivaCT 40 (Scanco Medical) was employed to scan a region of 20 mm in length covering the right paw and tarsal bones at a voltage of 70 keV with a current of 114 μA. For ex vivo microCT measurement, the bilateral hind paws were dissected for microCT measurement. After automatic reconstruction, 2D slices with 21 μm isotropic resolution were generated and used to select the region of interest (ROI) for 3D reconstruction. The whole bone tissue of calcaneus was subjected to quantitative evaluation with a threshold (sigma = 1.2, support = 2 and threshold = 200). Bone volume fraction (BV/TV) and bone mineral density (BMD) were calculated using built-in software[66].

**Bone histomorphometric analysis**. The bilateral hind paws were fixed in 70% ethanol. The whole skeletal tissues from bilateral hind paws were dehydrated in graded concentrations of ethanol and embedded without decalcification in modified methyl methacrylate. Frontal sections of the whole bone from the hind paws at a thickness of 10 μm was obtained using an EXAKT Cut/grinding System (EXAKT Technologies, Inc. Germany). Thereafter, the bone sections underwent bone histomorphometric analyses using professional image analysis software (BIOQUANT OSTEO analysis software) under fluorescence microscope (Leica image analysis system, Q500MC). The bone resorption parameters, *i.e.* Oc.S/BS and Oc.N/BS, were calculated and expressed according to standardized nomenclature for bone histomorphometry[67].

**DARTS assay**. Thirty microliter Flag-tagged AHR, AHR mutants and negative control BSA in 100 μL PBS was incubated with vehicle (DMSO) and Leflunomide at the indicated concentrations (5.0 and 10.0 μM) for 18 h at 4 °C, respectively. The samples were then digested with subtilisin (Sigma-Aldrich) at room temperature for 20 min. The reactions were stopped by adding SDS loading buffer and boiling for 5 min. Samples were loaded onto a 12% acrylamide SDS-PAGE gel and then stained with coomassie brilliant blue to visualize the banding pattern[27].

**Pull-down assay and LC-MS/MS**. To determine the direct interaction of AHR with Leflunomide, 30.0 μg Flag-tagged AHR and AHR mutants in 100 μL PBS was incubated with Leflunomide (10.0 μM), A77 1726 (10.0 μM) and vehicle (DMSO) for 18 h at 4 °C, respectively. Flag-tagged proteins were collected by a FLAG® M purification kit (Sigma-Aldrich). The small molecules in pull-down assays were extracted by 90% methanol (Sigma-Aldrich) and analyzed by Agilent 6400 Ultra

High Performance Liquid Chromatograph with Tripe Qquadrupole Mass Spectrometer (UHPLC-QqQ MS/MS) (Agilent Technologies)[28,68].

**Real-time PCR.** A RNeasy Mini Kit (Qiagen) was used to extract total RNA. Total RNA was reverse transcribed into cDNA using a QuantiTect Reverse Transcription Kit (Qiagen). The 10 μL volume of the final real-time PCR solution contained 1 μL diluted cDNA product, 5 μL 2 × Power SYBR® Green PCR Master Mix (Applied Biosystems), 0.5 μL each of forward and reverse primers and 3 μL nuclease-free water. The forward and reverse primers were shown as the following: *CRP*: 5′-ATGGAGAAGCTACTCTGGTGC -3′ (sense) and 5′-ACACACAGTAAAGGT GTTCAGTG-3′ (antisense); *TRAP5b*: 5′-GCAACATCCCCTGGTATGTG-3′ (sense) and 5′-GCAAACGGTAGTAAGGGCTG-3′ (antisense); *HIF1α*: 5′-TCCAT TATGAGGCTGACCATC-3′ (sense) and 5′-CCATCCTCAGAAAGCACCATA-3′ (antisense); *GAPDH*: 5′-AGGTCGGTGTGAACGGATTTG-3′ (sense) and 5′-TG TAGACCATGTAGTTGAGGTCA-3′ (antisense). The fluorescence signal emitted was collected by an ABI PRISM® 7900HT Sequence Detection System and the signal was converted into numerical values by SDS 2.1 software (Applied Biosystems)[66,69].

**Statistical analysis.** The binary logistic regression was used to determine the indication accuracies of blood biomedical indicators in PBE + RA patients. The Pearson's correlation was used to determine correlations between variables. The Shapiro-Wilk W test was used to test for normality of all continuous variables. When comparing measured outcomes in two independent groups, two-sided *t*-test was used for analyzing continuous variables. When comparing repeated measured outcomes in time series groups, repeated-measures analysis of variance (ANOVA) was used for analyzing the data. For difference among multiple independent groups, one-way ANOVA with a *post-hoc* test was performed. Samples were randomly located into different groups. The investigators were blinder to group allocation during data collection and analysis. A two-sided *P*-value of less than 0.05 was considered statistically significant. All statistical analyses were performed with a SPSS software, version 22.0. Two types of replicates including biological replicates and technical replicates was performed to yield accurate and reliable statistics. Sample size was calculated by the formula: $n = 2 [(U_\alpha + U_\beta) S \delta^{-1}]^2$. S in the formula referred to the standard deviation (s.d.). $\delta$ in the formula represented the inter-group difference of the mean values. There was a difference above 30 mg cm$^{-3}$ in BMD between treatment group and control group, *i.e.*, $\delta = 30$ mg cm$^{-3}$. s.d. of BMD from previous studies was 15 mg cm$^{-3}$, i.e., $S = 15$ mg cm$^{-3}$. We selected the significant level at 5% ($\alpha = 0.05$) in a two-tailed test and power of the study at 90% (1- $\beta = 0.9$). According to the formula: $n = 2 [(U_\alpha + U_\beta) S \delta^{-1}]^2$, $U_{0.05} = 1.960$, $\beta = 0.10$, $U_{0.1} = 1.282$, we chose $n = 9$ for each group, which was enough to detect the real difference among the treatment groups. Animals in poor body condition, such as tumors or other sick conditions during aging, were excluded. The exclusion was made before random group assignments, experimental intervention and data analysis.

**Reporting summary.** Further information on research design is available in the Nature Research Reporting Summary linked to this article.

## Data availability
The authors declare that all data supporting the findings of this study are available within the article and its Supplementary Information files or are available from the authors upon reasonable request. The source data underlying Figs. 1–6, Supplementary Fig. 1–4 and Supplementary Figs. 6–10 are provided as a Source Data file.

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

## Acknowledgements

We thank technical staffs (Ms. Yeuk Siu Cheung and Mr. Chi Leung Chan) from Law Sau Fai Institute for Advancing Translational Medicine in Bone and Joint Diseases, Hong Kong Baptist University for providing technical support. This work was supported by the Ministry of Science and Technology of China (2013ZX09301307 to A.L.), the Hong Kong General Research Fund (HKBU479111 to G.Z., HKBU478312 to G.Z., HKBU262913 to G.Z., HKBU12102914 to G.Z., HKBU261113 to A.L., 12101018 to F.L., CUHK14112915 to B-T.Z. and CUHK489213 to B-T.Z.), the RC's Start-up Grant for New Academics (162498 to C.L.) the Natural Science Foundation Council of China (81272045 to G.Z., 81700780 to C.L., 81703049 to F.L. and 81470072 to X.H.), the Research Grants Council and Natural Science Foundation Council of China (N_HKBU435/12 to G.Z.), the Croucher Foundation (Gnt#CAS14BU/CAS14201 to A.L.), the Interdisciplinary Research Matching Scheme (IRMS) of Hong Kong Baptist University (RC-IRMS/12-13/02 to A.L. and RC-IRMS/13-14/02 to G.Z.), the Hong Kong Baptist University Strategic Development Fund (SDF13-1209-P01 to A.L. and SDF15-0324-P02(b) to A.L.), the Hong Kong Research Grants Council Early Career Scheme (489213 to G.Z.), the Inter-institutional Collaborative Research Scheme of Hong Kong Baptist University (RC-ICRS/14-15/01 to G.Z. and RC-ICRS/16-17/01 to A.L.), the Faculty Research Grant of Hong Kong Baptist University (FRG1/13-14/024 to G.Z., FRG2/13-14/006 to G.Z. and FRG2/14-15/010 to G.Z.), the Science and Technology Innovation Commission of Shenzhen Municipality Funds (JCYJ20170307161659648 to F.L.), the China Academy of Chinese Medical Sciences (Z0252 and Z0293 to A.L.) and the National Key R&D Program of China (2018YFC1705205 to A.L. and 2018YFA0800804 to G.Z.).

## Author contributions

A.L., G.Z., B-T.Z., and C.L. jointly supervised the whole project. C.L., J.L., and C. Lu. performed the major research and wrote the paper in equal contribution. D. X., R.D., C.Z., H.Z., D.G., B.G., J.L. and B. H. provided the technical support. X.H., F. L. and W.Z. provided their professional expertise.

## Additional information

**Competing interests:** The authors declare no competing interests.

