## [Peer Review File · Nature Communications]

Reviewers' Comments:

Reviewer #1:

Remarks to the Author:

This manuscript reports an interesting finding that circulating C-reactive protein (CRP) might be used as an index to stratify RA patients for responsiveness to Leflunomide. The authors further examined the mechanisms whereby CRP might get involved. Though how Leflunomide regulates CRP expression appears to be convincingly demonstrated, why CRP is detrimental in RA remains elusive.

Specific comments:

- (1) Transgenic expression of human or rabbit CRP in mice has been shown to inhibit the development of CIA (Arthritis Rheum, 2011; Scand J Rheumatol, 2006), while knockout of CRP exacerbates CIA (Arthritis Rheum, 2011) and inflammation-induced bone damage (Front Immunol, 2018) in mice. Therefore, the protective role of CRP in CIA is seemingly incompatible with the detrimental role of CRP in CIA proposed in this manuscript. Of note, the only piece of direct in vivo evidence demonstrating the detrimental role of CRP in this manuscript is the treatment of CIA rats with anti-CRP antibodies. Moreover, it is unclear whether these antibodies act via reducing CRP levels, via forming immune complexes to regulate complement, or via other bystander effects.
- (2) IL-6 is the chief inducer of CRP expression in hepatocytes. The authors found that Leflunomide inhibited CRP expression in hepatocytes (Figure 3d), but this inhibition could be abrogated by IL-6 (Figure 3e). On the other hand, CRP was induced in both PBE+ and PBE- patients (over 10 µg/ml, denoting the presence of notable inflammation in the clinical practice; Figure 1e) but with comparable levels of upregulated IL-6 (Figure s1c). Therefore, it might be difficult to explain how Leflunomide can inhibit CRP expression in PBE- patients in the presence of upregulated IL-6.
- (3) The large and prevalent variations of circulating CRP levels found in inbreeding rats and mice (Figure 2c and Figure s8d) are somewhat surprising. In addition, the high levels of baseline CRP in mice (20-30 µg/ml) appear to be inconsistent to previous findings.
- (4) In anti-CRP treated CIA rats, non-CRP antibodies might be better control. It is also unclear why the authors did not examine the effects of anti-CRP antibodies in CRPL rats.
- (5) The authors propose that CRP promotes osteoclastic activities (Figure 3g). However, no evidence for this proposal has been provided in the manuscript, and previous study have identified no (Front Immunol, 2018) or controversial actions of CRP on osteoclastic activities (Arthritis Res Ther, 2015; Life Sci, 2016). Indeed, monomeric CRP instead of CRP has been reported to be active in regulating osteoclastic activities. Moreover, monomeric CRP appears to inhibit osteoclastic activities by neutralizing RANKL (Front Immunol, 2018).
- (6) The authors do not provide actual levels of CRP released by hepatocytes (Figure 3d, 3e and 4k). It is thus unclear whether these levels are relevant to in vivo conditions. Usually, the levels of CRP in cell culture are much lower than that in circulation. If it was that case, how will the authors relate the in vitro HIF modulation by CRP in overexpression and knockdown experiments to in vivo conditions.
- (7) Circulating levels of CRP in HIF KO mice should be provided and compared to that in control mice.
- (8) The authors' conclusion would predict that CRP KO would abrogate the differential responsiveness to Leflunomide in mice and rats. These experiments might be essential to support their conclusion.

Reviewer #2:

Remarks to the Author:

This manuscript explored the reasons why Leflunomide only ameliorates the bone erosion in partial patients with rheumatoid arthritis. Based on the analysis of clinical data and animal experiments, authors claimed that the levels of CRP and consequent HIF-1α play crucial roles in the sensitivity of patients to the Leflunomide therapy. This study includes interesting data, and the manuscript was well prepared. However, there are some issues to be considered.

1. It is unreasonable to divide CIA rats and mice into CRPH and CRPL based on the serum levels of CRP. In this study, Leflunomide or A77 1726 was orally administered from the day of first immunization. Partial animals, especially SD rats (the occurrence rates of arthritis are only 60%~70%), in model and treatment groups did not exhibit significant arthritis symptom. The lower level of CRP in these animals was due to the unsuccessfulness of model preparation, distinct from CRPL patients.
2. It is generally accepted that Leflunomide functions through its metabolite A77 1726. In this study, authors stated that Leflunomide itself but not A77 1726 could activate AhR to inhibit the expression of CRP in liver. More solid evidence is needed to support this conclusion. e.g., to give Leflunomide together with specific inhibitors of its metabolism.
3. AhR might function through genomic or nongenomic pathway. The latter pathway is independent on ARNT. AhR-ARNT binding/interaction might not represent the activation of AhR signaling.
4. During CIA, some pro-inflammatory cytokines such as IL-17, IL-6, and RANKL also play innegligible roles in the occurrence of bone erosion. Leflunomide might exerts protective effects on bone and cartilage by down-regulating the levels of these cytokines.

Reviewer #3:

Remarks to the Author:

In this paper, Liang and colleagues report that inhibition of HIF1alpha activity attenuates bone erosions in experimental models of Rheumatoid Arthritis (RA) characterized by high levels of C-reactive protein (CRP) and pharmacologically treated with leflunomide. The attenuation of bone erosions correlates with decreased levels of CRP. Of note, leflunomide treatment alone corrects inflammation but does not reduce bone erosions in the same experimental models. The authors take advantage of both genetic and pharmacological approaches to test their working hypothesis.

The study is interesting, novel and potentially important. However, a few issues need to be addressed to strengthen the authors' conclusions and their biological relevance.

Specific Comments:

1. Figure 2: Histological analysis of treated and untreated specimens with either high or low CRP should be performed to document synovial hyperplasia/inflammation and bone erosions. Along those lines, TRAP stained sections should be shown.
2. Figure 4: In vitro incubation of THLE-2 cells in hypoxia should be pursued as a positive control for HIF-1alpha stabilization and activity.
3. Figure 4: The modalities that lead to increased HIF-1alpha stabilization downstream of CRP are completely elusive. It would be helpful if the authors could more extensively discuss this point.
4. Figure 5: Histological analysis of control and mutant specimens with or without leflunomide treatment should be performed to document synovial hyperplasia/inflammation and bone erosions. Moreover, the authors should document efficient recombination of the HIF-1alpha floxed allele.
5. Figure 6: as above, in vitro incubation of THLE-2 cells in hypoxia should be pursued as positive control for HIF-1alpha stabilization and activity. Moreover, the authors should document that ACF treatment impairs HIF-1alpha activity by studying classical downstream targets of HIF-1alpha.
6. Figure 7: as above, histological analysis should be pursued to document synovial hyperplasia/inflammation and bone erosions.
7. Numerous typos are present and should be corrected.

Reviewer #4:

Remarks to the Author:

This is a well perfumed and beautiful studies addressing the mechanisms of Leflunamid for treatment in rheumatoid arthritis, RA. It is known that leflunamid is effecting for treatment of

inflammation but less effective for treatment of bone erosions, a common problem in severe RA. They show by combined experimental animals and clinical studies that the mechanisms is related to the level and MOA of CRP and suggest a way to overcome the lack of efficiency against bone erosions for clinical treatment. It is an unusual study combining both experimental insight and clinical practice. My expertise is more animal models than clinical studies and from my perspective the animal experiments are well performed, they have even taken care of using the relevant MHC class II molecules in the mouse models leading to that they actually use a proper CIA model for RA and not the less defined arthritis induced by collagen in B6 mice.

Point-by-point responses to reviewer #1' comments

Overall comment: This manuscript reports an interesting finding that circulating C-reactive protein (CRP) might be used as an index to stratify RA patients for responsiveness to Leflunomide. The authors further examined the mechanisms whereby CRP might get involved. Though how Leflunomide regulates CRP expression appears to be convincingly demonstrated, why CRP is detrimental in RA remains elusive.

Our response: Thanks for the positive and constructive comments. We understood the reviewer 1's concern regarding the detrimental role of CRP demonstrated in our manuscript. Actually, the exact role of CRP in RA has been an unsettled dispute for a long time. No human study to date has directly investigated the contribution of CRP in RA, and animal studies performed so far have shown mixed results. As mentioned by the reviewer 1 in the following specific comment 1, transgenic expression of human and rabbit CRP in mice has been shown to inhibit CIA development^{1, 2}, while knockout of CRP exacerbates CIA and bone damage^{1, 3}, suggesting that CRP plays a protective role in RA. However, CRP has also been reported to induce prolonged inflammation in arthritic rabbits and directly promotes osteoclastogenesis^{4, 5}, indicating that CRP makes a detrimental contribution to RA.

In our study, we also demonstrated the detrimental function of CRP by inhibition of CRP with anti-CRP antibodies in CIA rats. To validate this result in our revised manuscript, we used an *in vivo* RNA interference (RNAi)-based strategy to silence CRP expression in hepatocytes. We are experienced in targeted delivery of therapeutic siRNAs^{6, 7}. Lipid nanoparticles (LNPs) has been proven as liver-targeted delivery systems for siRNAs *in vivo*^{8, 9, 10, 11}. The relative success of LNPs for liver delivery of siRNAs can, at least in part, be attributed to the favorable physiology of the liver, a well-perfused organ with a fenestrated endothelium⁸. On the other hand, LNPs are known to interact with serum proteins that can direct them to hepatocytes after systemic administration¹¹. We developed the LNPs-based delivery systems for CRP siRNA according to the optimized formulation⁹. The CRP^H CIA rats were intravenously administrated with PBS (Veh), LNPs, LNPs encapsulating negative control siRNA (LNPs-NC siRNA) or LNPs encapsulating CRP siRNA (LNPs-CRP siRNA). About 90% of the administered dose of LNPs-CRP siRNA was found in liver, which was consistent with the results from a previous study⁹. **Please refer to the Supplementary Table 3 and the highlighted revision in Results (line 139-152) and Materials and Methods (line 493-504).** LNPs-CRP siRNA lowered CRP expression, attenuated bone erosion and prevented bone loss in CRP^H rats when compared to PBS, LNPs or LNPs-NC siRNA. **Please refer to the Supplementary Figure 3e-3i and the highlighted revision in Results (line 139-152) and Materials and Methods (line 493-504).** These results together with our previous data confirmed that CRP played a detrimental role in RA.

We compare the above studies which seem to draw incompatible conclusions about the role of CRP in RA. Regarding the studies demonstrating the protective role of CRP in RA^{1, 2, 3}, CRP expression is manipulated prior to arthritic induction in genetic models (CRP transgenic and knockout mice). For our work and another studies demonstrating the detrimental role of CRP in RA⁴, however, CRP level is influenced after arthritic induction in CIA models. The above studies actually reveal the different roles of CRP in distinct stages of RA development (early induction stage versus late active stage). Together, they suggest that CRP may have double-faced functions in RA. CRP may confer benefit during the early induction phase of RA but play a detrimental role during active RA, which is consistent with the assumption in one of the above study¹. **Please refer to the highlighted revision in Discussion (line 298-313).**

Specific comment 1: Transgenic expression of human or rabbit CRP in mice has been shown to inhibit the development of CIA (Arthritis Rheum, 2011; Scand J Rheumatol, 2006), while knockout of CRP exacerbates CIA (Arthritis Rheum, 2011) and inflammation-induced bone damage (Front Immunol, 2018) in mice. Therefore, the protective role of CRP in CIA is seemingly incompatible with the detrimental role of CRP in CIA proposed in this manuscript. Of note, the only piece of direct *in vivo* evidence demonstrating the detrimental role of CRP in this manuscript is the treatment of CIA rats with anti-CRP antibodies. Moreover, it is unclear whether these antibodies act via reducing CRP levels, via forming immune complexes to regulate complement, or via other bystander effects.

Our response: Thanks for the constructive comments. Actually, the exact role of CRP in RA has been an unsettled dispute for a long time. No human study to date has directly investigated the contribution of CRP in RA, and animal studies performed so far have shown mixed results. As mentioned by the reviewer 1 in this specific comment 1, transgenic expression of human and rabbit CRP in mice has been shown to inhibit CIA development^{1, 2}, while knockout of CRP exacerbates CIA and bone damage^{1, 3}, suggesting that CRP plays a protective role in RA. However, CRP has also been reported to induce prolonged inflammation in arthritic rabbits and directly promotes osteoclastogenesis^{4, 5}, indicating that CRP makes a detrimental contribution to RA.

In our study, we also demonstrated the detrimental function of CRP by inhibition of CRP with anti-CRP antibodies in CIA rats. To validate this result in our revised manuscript, we used an *in vivo* RNA interference (RNAi)-based strategy to silence CRP expression in hepatocytes. We are experienced in targeted delivery of therapeutic siRNAs^{6, 7}. Lipid nanoparticles (LNPs) has been proven as liver-targeted delivery systems for siRNAs *in vivo*^{8, 9, 10, 11}. The relative success of LNPs for liver delivery of siRNAs can, at least in part, be attributed to the favorable physiology of the liver, a well-perfused organ with a fenestrated endothelium⁸. On the other hand, LNPs are known to interact with serum proteins that can direct them to hepatocytes after systemic administration¹¹. We developed the LNPs-based delivery systems for CRP siRNA according to the optimized

formulation⁹. The CRP^H CIA rats were intravenously administrated with PBS (Veh), LNPs, LNPs encapsulating negative control siRNA (LNPs-NC siRNA) or LNPs encapsulating CRP siRNA (LNPs-CRP siRNA). About 90% of the administered dose of LNPs-CRP siRNA was found in liver, which was consistent with the results from a previous study⁹. **Please refer to the Supplementary Table 3 and the highlighted revision in Results (line 139-152) and Materials and Methods (line 493-504).** LNPs-CRP siRNA lowered CRP expression, attenuated bone erosion and prevented bone loss in CRP^H rats when compared to PBS, LNPs or LNPs-NC siRNA. **Please refer to the Supplementary Figure 3e-3i and the highlighted revision in Results (line 139-152) and Materials and Methods (line 493-504).** These results together with our previous data confirmed that CRP played a detrimental role in RA.

We compare the above studies which seem to draw incompatible conclusions about the role of CRP in RA. Regarding the studies demonstrating the protective role of CRP in RA^{1, 2, 3}, CRP expression is manipulated prior to arthritic induction in genetic models (CRP transgenic and knockout mice). For our work and another studies demonstrating the detrimental role of CRP in RA⁴, however, CRP level is influenced after arthritic induction in CIA models. The above studies actually reveal the different roles of CRP in distinct stages of RA development (early induction stage versus late active stage). Together, they suggest that CRP may have double-faced functions in RA. CRP may confer benefit during the early induction phase of RA but play a detrimental role during active RA, which is consistent with the assumption in one of the above study¹. **Please refer to the highlighted revision in Discussion (line 298-313).**

Specific comment 2: IL-6 is the chief inducer of CRP expression in hepatocytes. The authors found that Leflunomide inhibited CRP expression in hepatocytes (Figure 3d), but this inhibition could be abrogated by IL-6 (Figure 3e). On the other hand, CRP was induced in both PBE+ and PBE- patients (over 10 µg/ml, denoting the presence of notable inflammation in the clinical practice; Figure 1e) but with comparable levels of upregulated IL-6 (Figure s1c). Therefore, it might be difficult to explain how Leflunomide can inhibit CRP expression in PBE- patients in the presence of upregulated IL-6.

Our response: Thanks for the constructive comments. In our study, we found that PBE+ patients demonstrated a higher CRP level, while CRP level was relatively lower in PBE- patients when compared to PBE+ patients. Thus, we termed the PBE- individuals as “CRP^L” and the PBE+ individuals as “CRP^H” (Figure 1e). Inhibition of CRP attenuated bone erosion in CIA rats, suggesting the detrimental role of CRP in RA. We mainly focused on exploring the downstream signaling of high CRP-mediated failure of Leflunomide in attenuating bone erosion in CRP^H RA. We induced overexpression of CRP in hepatocytes *in vitro* by incubating them with IL-6, as IL-6 axis was the chief inducer of CRP expression in hepatocytes¹². We found that Leflunomide induced AHR activation to inhibit hepatic CRP expression in normal hepatocytes but not in hepatocytes

pretreated with IL-6 *in vitro*. Knockdown of CRP in hepatocytes pretreated with IL-6 recovered the efficacy of Leflunomide in induction of AHR activation, suggesting that IL-6 abrogated the Leflunomide-AHR-CRP signaling via induction of CRP overexpression. We further showed that overexpression of CRP upregulated HIF1 α , which competed with AHR for ARNT association and interfered Leflunomide-AHR-CRP signaling, leading to the limited efficacy of Leflunomide in PBE+ (CRP^H) RA individuals. Even though the PBE- (CRP^L) RA individuals had a relatively lower serum CRP when compared to the higher CRP in the PBE+ (CRP^H) individuals, CRP in PBE- individuals was mildly elevated in contrast to the normal reference range in clinical practice¹³. Our results showed that Leflunomide could inhibit CRP expression in PBE- (CRP^L) individuals rather than in PBE+ (CRP^H) individuals (Figure 1f, Figure 2d and Supplementary Figure 9e), implying that the mildly elevated CRP in CRP^L individuals might be insufficient for interfering the Leflunomide-AHR-CRP signaling *in vivo*, while the higher CRP in CRP^H individuals could significantly interfere the signaling. To validate this speculation, we further isolated hepatocytes from CRP^L and CRP^H CIA rats treated with Leflunomide or vehicle. Binding of AHR with ARNT and CRP inhibition induced by Leflunomide was remarkable in hepatocytes from CRP^L CIA rats but not significant in hepatocytes from CRP^H CIA rats. **Please refer to the Supplementary Figure 8e and 8f and the highlighted revision in Results (line 226-228) and Material and Methods (line 451-456).**

As mentioned by the reviewer 1, serum CRP level greater than 10 mg/l is a sign of chronic inflammation in clinical practice¹³. Our data showed that both PBE+ (CRP^H) and PBE- (CRP^L) RA patients had serum CRP levels above 10 mg/l. We investigated the mechanism underlying the distinctive CRP levels between the two groups of RA individuals. IL-6 axis was the chief inducer of CRP expression in hepatocytes¹². We found that serum IL-6 were comparable between PBE+ and PBE- individuals (Supplementary Figure 1a, Supplementary Figure 2c and Supplementary Figure 9c), while IL-6 receptor (IL-6R) in liver was differentially expressed between the two groups. **Please refer to the Supplementary Figure 2e and the highlighted revision in Results (line 132-134).** This finding was interesting. We will perform big data analysis to confirm this finding and reveal the difference of upstream signaling network of CRP expression between the two groups of RA individuals.

Specific comment 3: The large and prevalent variations of circulating CRP levels found in inbreeding rats and mice (Figure 2c and Figure s8d) are somewhat surprising. In addition, the high levels of baseline CRP in mice (20-30 μ g/ml) appear to be inconsistent to previous findings.

Our response: Understood the reviewer 1's concern. In our study, Sprague-Dawley (SD) rats and DBA/1 mice were used for establishment of CIA, as they are common animal models for exploring the pathogenic mechanism of RA and therapeutic action of anti-arthritic agents^{14, 15}. SD rat is an outbred strain and DBA/1 mouse is an inbred strain^{16, 17}. We find the large variations of serum CRP

levels in both SD rats and DBA/1 mice, which are the basis for classification of CIA models into CRP^L and CRP^H subgroups (Figure 2c and Supplementary Figure 9d). For many decades, inbred rats and mice are preferred over outbred strains because it is assumed that they display lower levels of variations due to their greater homogeneity¹⁸. However, very few studies with adequate sample sizes have in fact compared phenotypic variations between inbred strains and outbred strains of animals¹⁹. Surprisingly, recent reports suggest that inbred and outbred animals show comparable phenotypic variations^{18, 19, 20}. On one hand, they conclude that the outbred animals could be widely adopted as research subjects like the inbred animals. On the other hand, it could not be ignored that the coefficients of variation vary in large ranges for both of them¹⁸. We reviewed other studies, which also showed large variations of serum CRP^{21, 22}. In our previous work, we also demonstrate the large variations of other proteins in signaling pathways²³. Even though the large variations of proteins in established animal models seem to differ from the common sense and the underlying reasons are still unknown, the variations could be used for explaining non-response of therapeutics in human disease²³. **Please refer to the highlighted revision in Discussion (line 341-355).**

Previously, pertinent publications frequently reiterate that mouse CRP is a trace protein and serum levels of CRP in mice are markedly lower than in humans and rabbits, < 3 mg/ml even after inflammatory stimuli^{2, 24}. However, recent studies argue that such statements is not convincing as they either refer to review articles devoid of experimental data or an early publication, which fails to detect any mouse CRP using sheep anti-human CRP antibodies cross-reacting with mouse CRP before acute phase induction^{25, 26}. To date, reliable reagents specific for mouse have become available and studies suggest that serum CRP levels may be vastly underestimated^{26, 27}. High serum CRP levels ranging from 5 mg/ml to 50 mg/ml have been determined in different mouse models^{21, 22, 26, 28, 29, 30}, which are consistent with our results. **Please refer to the highlighted revision in Discussion (line 332-339).**

Specific comment 4: In anti-CRP treated CIA rats, non-CRP antibodies might be better control. It is also unclear why the authors did not examine the effects of anti-CRP antibodies in CRPL rats.

Our response: Thanks for the constructive comments. IgG isotypes are usually used as the negative controls for *in vivo* antibody therapy^{31, 32}. In our revised manuscript, we used rabbit IgG as the negative controls when examining the effects of rabbit anti-rat CRP antibodies on bone erosion in CRP^H CIA rats. Our results showed that anti-CRP antibodies effectively attenuated bone erosion and prevented bone loss, while IgG had no therapeutic effects. **Please refer to the Supplementary Figure 3a-3d and the highlighted revision in Results (line 137-129) and Materials and Methods (line 544-545).**

Previously, we did not examine the effects of anti-CRP antibodies in CRP^L CIA rats. Reasons are as follows. We began this study based on our observation that Leflunomide was effective in attenuating bone erosion in CRP^L RA individuals but showed limited efficacy in CRP^H individuals. This finding driven us to propose an initial idea about developing a precision medicine-based therapeutic strategy for RA. Thus, we focused on exploring an alternative treatment option for the CRP^H RA individuals who poorly responded to Leflunomide. Previous reports suggest that CRP stimulates osteoclastogenesis and plays a detrimental role during active RA^{4, 5}. Serum CRP was positively associated with TRAP5b in CRP^H RA patients (Figure 1g), implying that CRP inhibition could be a therapeutic strategy for attenuating bone erosion in CRP^H RA. Thus, we examined the effects of anti-CRP antibodies on bone erosion in CRP^H CIA rats but not in CRP^L CIA rats as the CRP^L CIA rats showed satisfactory responsiveness to Leflunomide.

In our revised manuscript, we further tested whether anti-CRP antibodies could attenuate bone erosion in CRP^L CIA rats to address the reviewer 1's question. As we have proven that besides immunomodulation via metabolite A77 1726, Leflunomide itself could induce AHR activation to inhibit hepatic CRP production and then attenuate bone erosion in CRP^L arthritic rats (Figure 3g). We speculated that CRP inhibition by anti-CRP antibodies might partially mimic the therapeutic effects of Leflunomide in CRP^L CIA rats. As expected, the anti-CRP antibodies effectively attenuated bone resorption and bone erosion in CRP^L rats when compared to PBS or IgG controls. **Please refer to the Supplementary Figure 4e-4h and the highlighted revision in Results (line 151-152).**

Specific comment 5: The authors propose that CRP promotes osteoclastic activities (Figure 3g). However, no evidence for this proposal has been provided in the manuscript, and previous study have identified no (Front Immunol, 2018) or controversial actions of CRP on osteoclastic activities (Arthritis Res Ther, 2015; Life Sci, 2016). Indeed, monomeric CRP instead of CRP has been reported to be active in regulating osteoclastic activities. Moreover, monomeric CRP appears to inhibit osteoclastic activities by neutralizing RANKL (Front Immunol, 2018).

Our response: Understood the reviewer 1's concern. Previously, Kim et al. report that CRP promotes osteoclastic activities via FcγRs signaling with the absence of RANKL⁵, while Cho et al. demonstrate that CRP inhibits RANKL-induced osteoclastic activities *in vitro*³³. Recently, Jia et al. suggest that CRP regulates osteoclast differentiation in conformation- and RANKL dependent manners³. Circulating native CRP is composed of five identical subunits but dissociates into the monomeric conformation upon entering local lesions³⁴. Jia et al. show that monomeric CRP induces osteoclast differentiation with the absence of RANKL *in vitro*³, which is consistent with Kim's report⁵. However, Jia et al. also demonstrate that monomeric CRP inhibits RANKL-induced osteoclastic differentiation by neutralizing RANKL *in vitro*³, which is consistent and accounts for Cho's findings³³.

Jia et al. further explain that native CRP is inefficient for osteoclastic activity *in vitro* but the prolonged incubation time (3-21 days) in cell culture would favor the conversion of native CRP to monomeric CRP³, suggesting that in both Kim and Cho's studies, CRP regulates osteoclastic activity actually via the monomeric conformation, as they incubate cells with CRP for 3-21 days and 3-5 days, respectively^{5, 33}. Based on these studies, we think whether CRP promotes or inhibits osteoclastic activities *in vivo* may be dependent on ratio of monomeric CRP to RANKL. We find that molar concentration of the monomeric CRP is over 10,000-fold of RANKL in both CRP^L and CRP^H RA patients, indicating that the monomeric CRP is enough to neutralizing RANKL and the redundant free monomeric CRP would dominate osteoclastic activities in RA. **Please refer to the Supplementary Figure 1d and the highlighted revision in Results and Materials and Methods (line 109-117 and line 486).**

Specific comment 6: The authors do not provide actual levels of CRP released by hepatocytes (Figure 3d, 3e and 4k). It is thus unclear whether these levels are relevant to *in vivo* conditions. Usually, the levels of CRP in cell culture are much lower than that in circulation. If it was that case, how will the authors relate the *in vitro* HIF modulation by CRP in overexpression and knockdown experiments to *in vivo* conditions.

Our response: Thanks for the constructive comments. As mentioned by the reviewer 1, we provided the relative levels of CRP released by hepatocytes rather than the actual CRP levels in cell culture experiments *in vitro* (Figure 3d, Figure 3f and Figure 4k). We had our own concerns as follows. In fact, the actual CRP levels released by hepatocytes in cell culture experiments *in vitro* usually rely on many anthropic conditions, such as medium volumes, cell densities, incubation times, sizes of culture plates, etc. It is difficult to perform all cell culture experiments using an absolutely identical conditions across different *in vitro* studies. Thus, we calculated relative levels of CRP in treatment groups by normalizing them to vehicle or controls in our cell culture experiments *in vitro*, thus making our manuscript clear and readable.

In our *in vitro* experiments, we found that overexpression of CRP increased HIF1 α level, whereas knockdown of CRP reduced expression of HIF1 α in hepatocytes, suggesting that hepatic CRP positively modulated HIF1 α expression *in vitro*. To relate these results with *in vivo* conditions, we examined levels of HIF1 α in hepatocytes from non-immunized, CRP^L and CRP^H CIA rats. Both CRP^L and CRP^H CIA rats had elevated HIF1 α expression than non-immunized rats. HIF1 α level was significantly higher in CRP^H CIA rats when compared to CRP^L CIA rats. There was a positive association between CRP and hepatic HIF1 α expression in both CRP^L and CRP^H CIA rats. **Please refer to Figure 4a and Supplementary Figure 8a and the highlighted revision in Results (line 214-218).** In addition, CRP inhibition by LNPs-CRP siRNA led to a reduction of HIF1 α expression in hepatocytes from CRP^H rats. **Please refer to Supplementary Figure 8b and the highlighted revision in Results (line 218-219).** These *in vivo* results together with our *in vitro* data,

demonstrated that CRP was an upstream molecule of HIF1 α and could positively regulate HIF1 α expression in hepatocytes.

Specific comment 7: Circulating levels of CRP in HIF KO mice should be provided and compared to that in control mice.

Our response: Agree with the reviewer 1's comments. In our revised manuscript, we examined whether hepatocyte-specific deletion of HIF1 α affected CRP expression *in vivo*. Our results showed that serum CRP level was comparable between the hepatocyte-specific HIF1 α knockout (HIF1 α -HKO) mice and the control HIF1 α ^{loxP/loxP} mice. **Please refer to the Supplementary Figure 10c and the highlighted revision in Results (250-252).**

Specific comment 8: The authors' conclusion would predict that CRP KO would abrogate the differential responsiveness to Leflunomide in mice and rats. These experiments might be essential to support their conclusion.

Our response: Agree with the reviewer 1's comments. In our revised manuscript, we determined whether CRP knockdown by liver-targeted LNPs-siRNA could abrogate the differential responsiveness to Leflunomide between CRP^H and CRP^L CIA rats. Our results showed that Leflunomide in combination with LNPs-siRNA effectively attenuated bone erosion in CRP^H CIA rats. The therapeutic efficacy of the combination in CRP^H CIA rats was equivalent to that of Leflunomide alone in CRP^L CIA rats. **Please refer to the Supplementary Figure 4a-4d and the highlighted revision in Results (line 146-150).**

Point-by-point responses to reviewer #2' comments

Overall comment: This manuscript explored the reasons why Leflunomide only ameliorates the bone erosion in partial patients with rheumatoid arthritis. Based on the analysis of clinical data and animal experiments, authors claimed that the levels of CRP and consequent HIF-1 α play crucial roles in the sensitivity of patients to the Leflunomide therapy. This study includes interesting data, and the manuscript was well prepared. However, there are some issues to be considered.

Our response: Thanks for the positive comments. Regarding the issues raised by the reviewer 2, we made the following point-by-point response.

Specific comment 1: It is un reasonable to divide CIA rats and mice into CRPH and CRPL based on the serum levels of CRP. In this study, Leflunomide or A77 1726 was orally administered from the day of first immunization. Partial animals, especially SD rats (the occurrence rates of arthritis are only 60%~70%), in model and treatment groups did not exhibit significant arthritis symptom. The lower level of CRP in these animals was due to the unsuccessfulness of model preparation, distinct from CRPL patients.

Our response: Understood the reviewer 2's concern. Actually, Leflunomide or A77 1726 was orally administered to rats and mice after immunization with type II collagen. **Please refer to Figure 2a, Figure 6a, Supplementary Figure 3a and 3e, Supplementary Figure 4a and 4e and Supplementary Figure 9a.** As mentioned by the reviewer 2, occurrence rate of collagen-induced arthritis in SD rats are 60%-70%³⁵. However, we excluded the unsuccessful animal models prior to dividing CRP^H and CRP^L CIA animals. We confirmed that all the CRP^H and CRP^L CIA animals exhibited arthritic symptom before drug treatment, as evidenced by microCT data and clinical arthritic score of baseline group in contrast to the non-immunized group. **Please refer to the Figure 2a, Figure 6b, Supplementary Figure 3b and 3g, Supplementary Figure 4b and 4f, Supplementary Figure 9b and Supplementary Figure 10d and 10e.**

Specific comment 2: It is generally accepted that Leflunomide functions through its metabolite A77 1726. In this study, authors stated that Leflunomide itself but not A77 1726 could activate AhR to inhibit the expression of CRP in liver. More solid evidence is needed to support this conclusion. e.g., to give Leflunomide together with specific inhibitors of its metabolism.

Our response: Thanks for the constructive comments. As mentioned by the reviewer 2, it is generally accepted that Leflunomide functions through its metabolite A77 1726³⁶. Previously, we treated human normal hepatocyte cell line (THLE-2) with DMSO (vehicle), Leflunomide and A77 1726, respectively. Our results showed that Leflunomide, rather than A77 1726, induced binding of AHR with ARNT (AHR activation) and decreased CRP expression in dose-dependent manners. After knockdown of AHR, Leflunomide could not decrease CRP level in THLE-2 (Figure 3c and 3d, Supplementary Figure 6c and 6d). These results suggest that Leflunomide itself but not A77 1726 activates AHR to inhibit CRP expression in normal hepatocytes. In our revised manuscript, we provided more solid evidences to support this conclusion based on the reviewer 2's advice. Leflunomide is mainly metabolized to A77 1726 by a cytochrome P450 1A2 enzyme (CYP1A2)³⁷. Furfurylline, a specific inhibitor of CYP1A2, could suppress metabolism of Leflunomide^{37, 38}. Human normal primary hepatocytes are considered as the gold standard model for drug metabolism studies^{39, 40, 41}. We incubated human normal primary hepatocytes with Leflunomide in the presence of Furfurylline or not. Leflunomide induced binding of AHR with ARNT and decreased CRP expression and release. Stabilization of Leflunomide via inhibition of its metabolism by Furfurylline

further enhanced AHR-ARNT interaction and CRP inhibition. **Please refer to Supplementary Figure 6e and 6f and the highlighted revision in Results (line 182-187).**

Specific comment 3: AhR might function through genomic or nongenomic pathway. The latter pathway is independent on ARNT. AhR-ARNT binding/interaction might not represent the activation of AhR signaling.

Our response: Thanks for the constructive comments. As mentioned by the reviewer 2, AHR functions through both genomic and nongenomic pathways⁴². The genomic pathway of AHR is already well-characterized and it is dependent on AHR-ARNT interaction. In addition to the genomic pathway, AHR also functions through nongenomic pathways, which is independent of AHR-ARNT interaction. Some ligands have been reported to trigger AHR nongenomic pathways to regulate activities of certain proteins, such as Src, JunD and cyclin A^{42, 43}. We have found that Leflunomide induces AHR-ARNT interaction to activate AHR genomic signaling. In this revised manuscript, we determined whether Leflunomide had any effects on non-genomic pathways of AHR. Our results demonstrated that Leflunomide had no effects on expression of JunD and cyclin A and Src activation. These results indicated that Leflunomide specifically induced ARNT-dependent AHR genomic signaling. **Please refer to Supplementary Figure 6g and the highlighted revision in Results (line 70-71, line 187-194, line 223 and line 234).**

Specific comment 4: During CIA, some pro-inflammatory cytokines such as IL-17, IL-6, and RANKLE also play innegligible roles in the occurrence of bone erosion. Leflunomide might exerts protective effects on bone and cartilage by down-regulating the levels of these cytokines.

Our response: Thanks for the constructive comments. As mentioned by the reviewer 2, cytokines (such as IL-17, IL-6 and RANKL) produced by immune cells and inflammatory synovial fibroblasts play critical roles in the occurrence of bone erosion in RA^{44, 45}. In our revised manuscript, we examined their serum levels in PBE- and PBE+ RA patients or CIA animal models after Leflunomide treatment. Our results showed that Leflunomide decreased serum IL-17, IL-6 and RANKL levels in both PBE- and PBE+ individuals. **Please refer to the Supplementary Figure 1a, Supplementary Figure 2c and Supplementary 9c and the highlighted revision in Results (line 98-100, line 124-127, line 261-243, line 482 and line 485-486).** These were consistent with our other results that the well-known immunomodulatory action of Leflunomide was significant in both PBE- and PBE+ RA individuals (Figure 1c, Figure 2b and Supplementary 9c).

Point-by-point responses to reviewer #3' comments

Overall comment: In this paper, Liang and colleagues report that inhibition of HIF1alpha activity attenuates bone erosions in experimental models of Rheumatoid Arthritis (RA) characterized by high levels of C-reactive protein (CRP) and pharmacologically treated with leflunomide. The attenuation of bone erosions correlates with decreased levels of CRP. Of note, leflunomide treatment alone corrects inflammation but does not reduce bone erosions in the same experimental models. The authors take advantage of both genetic and pharmacological approaches to test their working hypothesis. The study is interesting, novel and potentially important. However, a few issues need to be addressed to strengthen the authors' conclusions and their biological relevance.

Our response: Thanks for the positive comments. Regarding the few issues raised by the reviewer 2, we made the following point-by-point response.

Specific comment 1: Figure 2: Histological analysis of treated and untreated specimens with either high or low CRP should be performed to document synovial hyperplasia/inflammation and bone erosions. Along those lines, TRAP stained sections should be shown.

Our response: In our revised manuscript, we performed histological analysis to document inflammatory synovial hyperplasia and bone erosion. Hematoxylin and eosin (H&E) staining showed that Leflunomide attenuated inflammatory synovial hyperplasia in both the PBE- and PBE+ CIA rats. **Please refer to the Supplementary Figure 2d and the highlighted revision in Results (line 126-127).** These were consistent with our other results that the well-known immunomodulatory action of Leflunomide was significant in both PBE- and PBE+ CIA rats (Figure 2b). TRAP staining data demonstrated that Leflunomide inhibited osteoclastic activity in PBE- (CRP^L) CIA rats but not in PBE+ (CRP^H) CIA rats. **Please refer to the Supplementary Figure 2d the highlighted revision in Results (line 129-131).**

Specific comment 2: Figure 4: In vitro incubation of THLE-2 cells in hypoxia should be pursued as a positive control for HIF-1alpha stabilization and activity.

Our response: Thanks for the constructive comments. In our revised manuscript, THLE-2 cells incubated in hypoxia were used as a positive control. Hypoxia induced high expression of HIF1 α and increased binding of ARNT with HIF1 α in THLE-2 cells. **Please refer to the Figure 4d and 4e.**

Specific comment 3: Figure 4: The modalities that lead to increased HIF-1alpha stabilization downstream of CRP are completely elusive. It would be helpful if the authors could more extensively discuss this point.

Our response: Thanks for the constructive comments. In our *in vitro* experiments, we found that overexpression of CRP increased HIF1 α level, whereas knockdown of CRP reduced expression of HIF1 α in hepatocytes, suggesting that hepatic CRP positively modulated HIF1 α expression *in vitro*. To relate these results with *in vivo* conditions, we examined levels of HIF1 α in hepatocytes from non-immunized, CRP^L and CRP^H CIA rats. Both CRP^L and CRP^H CIA rats had elevated HIF1 α expression than non-immunized rats. HIF1 α level was significantly higher in CRP^H CIA rats when compared to CRP^L CIA rats. There was a positive association between CRP and hepatic HIF1 α expression in both CRP^L and CRP^H CIA rats. **Please refer to Figure 4a and Supplementary Figure 8a and the highlighted revision in Results (line 214-218).** In addition, CRP inhibition by LNPs-CRP siRNA led to a reduction of HIF1 α expression in hepatocytes from CRP^H rats. **Please refer to Supplementary Figure 8b and the highlighted revision in Results (line 218-219).** These *in vivo* results together with our *in vitro* data, demonstrated that CRP was an upstream molecule of HIF1 α and could positively regulate HIF1 α expression in hepatocytes.

Specific comment 4: Figure 5: Histological analysis of control and mutant specimens with or without leflunomide treatment should be performed to document synovial hyperplasia/inflammation and bone erosions. Moreover, the authors should document efficient recombination of the HIF-1alpha floxed allele.

Our response: Agree with the reviewer 3's comments. In our revised manuscript, we performed histological analysis to document inflammatory synovial hyperplasia and bone erosion in CRP^L and CRP^H subgroups of HIF1 α -HKO and control HIF1 α ^{loxP/loxP} CIA mice with or without Leflunomide treatment. H&E staining data showed that inhibition of inflammatory synovial hyperplasia by Leflunomide in CRP^L and CRP^H subgroups was comparable for both HIF1 α ^{loxP/loxP} and HIF1 α -HKO CIA rats. **Please refer to the Supplementary Figure 10f and the highlighted revision in Results (line 256-258).** TRAP staining demonstrated that Leflunomide decreased osteoclastic activity in CRP^L HIF1 α ^{loxP/loxP} CIA mice but not in CRP^H HIF1 α ^{loxP/loxP} CIA mice. However, Leflunomide inhibited osteoclastic activity in both CRP^L and CRP^H subgroups of HIF1 α -HKO CIA mice. **Please refer to the Supplementary Figure 10i and the highlighted revision in Results (line 258-261).**

Moreover, we demonstrated the efficient recombination of the HIF1 α floxed allele in HIF1 α ^{loxP/loxP} mice. The data was consistent with the genotyping report from Jackson Laboratory. **Please refer to the Supplementary Figure 10a and the highlighted revision in Results (line 245-250).** Deletion of HIF-1 α was observed in HIF1 α -HKO but not in HIF1 α ^{loxP/loxP} mice. **Please refer to the Supplementary Figure 10b and the highlighted revision in Results (line 245-250).**

Specific comment 5: Figure 6: as above, in vitro incubation of THLE-2 cells in hypoxia should be pursued as positive control for HIF-1alpha stabilization and activity. Moreover,

the authors should document that ACF treatment impairs HIF-1alpha activity by studying classical downstream targets of HIF-1alpha.

Our response: Thanks for the constructive comments. In our revised manuscript, THLE-2 cells incubated in hypoxia were used as a positive control. Hypoxia induced high expression of HIF1 α and increased binding of ARNT with HIF1 α in THLE-2 cells. ACF had no effects on HIF1 α expression but decreased the binding of ARNT with HIF1 α , which was consistent with previous reports that ACF was a selective inhibitor blocking HIF1-ARNT interaction^{46, 47, 48}. **Please refer to the Figure 5a and the highlighted revision in Results (line 266-268).** We also examined effects of ACF on the most classical target gene of HIF1 α -ARNT signaling, *i.e.*, vascular endothelial growth factor (VEGF)⁴⁹. Our results showed that ACF downregulated expression of VEGF in THLE-2 cells. **Please refer to the Figure 5a and the highlighted revision in Results (line 266-268).**

Specific comment 6: Figure 7: as above, histological analysis should be pursued to document synovial hyperplasia/inflammation and bone erosions.

Our response: Thanks for the constructive comments. In our revised manuscript, we performed histological analysis to document inflammatory synovial hyperplasia and bone erosion. H&E staining data showed that Leflunomide alone or the combination of Leflunomide with ACF attenuated inflammatory synovial hyperplasia in CRP^H CIA rats. **Please refer to the Supplementary Figure 11 and the highlighted revision in Results (line 285-287).** TRAP staining results demonstrated that the combination of Leflunomide and ACF inhibited bone resorption in CRP^H CIA rats. **Please refer to the Supplementary Figure 11 and the highlighted revision in Results (line 282-285).**

Specific comment 7: Numerous typos are present and should be corrected.

Our response: We appreciated the reviewer 3's patience for reading our manuscript with numerous typos. In the revised manuscript, we have carefully checked the grammar, words and figures and corrected the typos.

Point-by-point responses to reviewer #4' comments

Overall comment: This is a well perfumed and beautiful studies addressing the mechanisms of Leflunomide for treatment in rheumatoid arthritis, RA. It is known that leflunomide is effecting for treatment of inflammation but less effective for treatment of bone erosions, a common problem in

severe RA. They show by combined experimental animals and clinical studies that the mechanisms is related to the level and MOA of CRP and suggest a way to overcome the lack of efficiency against bone erosions for clinical treatment. It is an unusual study combining both experimental insight and clinical practice. My expertise is more animal models than clinical studies and from my perspective the animal experiments are well performed, they have even taken care of using the relevant MHC class II molecules in the mouse models leading to that they actually use a proper CIA model for RA and not the less defined arthritis induced by collagen in B6 mice.

Our response: Thanks for the positive comments.

References

1. Jones NR, *et al.* Collagen-induced arthritis is exacerbated in C-reactive protein-deficient mice. *Arthritis Rheum* **63**, 2641-2650 (2011).
2. Jiang S, Xia D, Samols D. Expression of rabbit C-reactive protein in transgenic mice inhibits development of antigen-induced arthritis. *Scand J Rheumatol* **35**, 351-355 (2006).
3. Jia ZK, *et al.* Monomeric C-Reactive Protein Binds and Neutralizes Receptor Activator of NF-kappaB Ligand-Induced Osteoclast Differentiation. *Front Immunol* **9**, 234 (2018).
4. Phillips NC. Exacerbation of experimental poly-D-lysine arthritis by C-reactive protein. *Agents Actions* **12**, 344-347 (1982).
5. Kim KW, Kim BM, Moon HW, Lee SH, Kim HR. Role of C-reactive protein in osteoclastogenesis in rheumatoid arthritis. *Arthritis Res Ther* **17**, 41 (2015).
6. Liang C, *et al.* Aptamer-functionalized lipid nanoparticles targeting osteoblasts as a novel RNA interference-based bone anabolic strategy. *Nat Med* **21**, 288-294 (2015).
7. Zhang G, *et al.* A delivery system targeting bone formation surfaces to facilitate RNAi-based anabolic therapy. *Nat Med* **18**, 307-314 (2012).
8. Akinc A, *et al.* Targeted delivery of RNAi therapeutics with endogenous and exogenous ligand-based mechanisms. *Mol Ther* **18**, 1357-1364 (2010).
9. Akinc A, *et al.* A combinatorial library of lipid-like materials for delivery of RNAi therapeutics. *Nat Biotechnol* **26**, 561-569 (2008).
10. Akinc A, *et al.* Development of lipidoid-siRNA formulations for systemic delivery to the liver. *Mol Ther* **17**, 872-879 (2009).
11. Semple SC, *et al.* Rational design of cationic lipids for siRNA delivery. *Nat Biotechnol* **28**, 172-176 (2010).
12. Schmidt-Arras D, Rose-John S. IL-6 pathway in the liver: From physiopathology to therapy. *J Hepatol* **64**, 1403-1415 (2016).
13. McBride JD, Cooper MA. A high sensitivity assay for the inflammatory marker C-Reactive protein employing acoustic biosensing. *J Nanobiotechnology* **6**, 5 (2008).
14. Brand DD, Latham KA, Rosloniec EF. Collagen-induced arthritis. *Nat Protoc* **2**, 1269-1275 (2007).
15. Trentham DE, Townes AS, Kang AH. Autoimmunity to type II collagen an experimental model of arthritis. *J Exp Med* **146**, 857-868 (1977).
16. Festing MF. Evidence should trump intuition by preferring inbred strains to outbred stocks in preclinical research. *ILAR J* **55**, 399-404 (2014).
17. Perez CJ, Dumas A, Vallieres L, Guenet JL, Benavides F. Several classical mouse inbred strains, including DBA/2, NOD/Lt, FVB/N, and SJL/J, carry a putative loss-of-function allele of Gpr84. *J Hered* **104**, 565-571 (2013).
18. Tuttle AH, Philip VM, Chesler EJ, Mogil JS. Comparing phenotypic variation between inbred and outbred mice. *Nat Methods* **15**, 994-996 (2018).
19. Jensen VS, Porsgaard T, Lykkesfeldt J, Hvid H. Rodent model choice has major impact on variability of standard preclinical readouts associated with diabetes and obesity research. *Am J Transl Res* **8**, 3574-3584 (2016).
20. Vaickus LJ, Bouchard J, Kim J, Natarajan S, Remick DG. Inbred and outbred mice have equivalent variability in a cockroach allergen-induced model of asthma. *Comp Med* **60**, 420-

- 426 (2010).
21. van den Berg S, *et al.* Distinctive cytokines as biomarkers predicting fatal outcome of severe *Staphylococcus aureus* bacteremia in mice. *PLoS One* **8**, e59107 (2013).
 22. Tseng HW, *et al.* Early anti-inflammatory intervention ameliorates axial disease in the proteoglycan-induced spondylitis mouse model of ankylosing spondylitis. *BMC Musculoskelet Disord* **18**, 228 (2017).
 23. Liang C, *et al.* Inhibition of osteoblastic Smurf1 promotes bone formation in mouse models of distinctive age-related osteoporosis. *Nat Commun* **9**, 3428 (2018).
 24. Pepys MB, Hirschfield GM. C-reactive protein: a critical update. *J Clin Invest* **111**, 1805-1812 (2003).
 25. Pepys MB. Isolation of serum amyloid P-component (protein SAP) in the mouse. *Immunology* **37**, 637-641 (1979).
 26. Teupser D, *et al.* No reduction of atherosclerosis in C-reactive protein (CRP)-deficient mice. *J Biol Chem* **286**, 6272-6279 (2011).
 27. Torzewski M, Waqar AB, Fan J. Animal models of C-reactive protein. *Mediators Inflamm* **2014**, 683598 (2014).
 28. Laucho-Contreras ME, Polverino F, Rojas-Quintero J, Wang X, Owen CA. Club cell protein 16 (Cc16) deficiency increases inflamm-aging in the lungs of mice. *Physiol Rep* **6**, e13797 (2018).
 29. Li Y, *et al.* Anti-inflammatory effects in a mouse osteoarthritis model of a mixture of glucosamine and chitooligosaccharides produced by bi-enzyme single-step hydrolysis. *Sci Rep* **8**, 5624 (2018).
 30. Ghosh C, Bishayi B. Characterization of Toll-like receptor-4 (TLR-4) in the spleen and thymus of Swiss albino mice and its modulation in experimental endotoxemia. *J Immunol Res* **2015**, 137981 (2015).
 31. Liang B, *et al.* Evaluation of anti-IL-6 monoclonal antibody therapy using murine type II collagen-induced arthritis. *J Inflamm (Lond)* **6**, 10 (2009).
 32. Yoshino S. Treatment with an anti-IL-4 monoclonal antibody blocks suppression of collagen-induced arthritis in mice by oral administration of type II collagen. *J Immunol* **160**, 3067-3071 (1998).
 33. Cho IJ, *et al.* Effects of C-reactive protein on bone cells. *Life Sci* **145**, 1-8 (2016).
 34. Wu Y, Potempa LA, El Kebir D, Filep JG. C-reactive protein and inflammation: conformational changes affect function. *Biol Chem* **396**, 1181-1197 (2015).
 35. Shi M, *et al.* The protective effects of chronic intermittent hypobaric hypoxia pretreatment against collagen-induced arthritis in rats. *J Inflamm (Lond)* **12**, 23 (2015).
 36. Herrmann ML, Schleyerbach R, Kirschbaum BJ. Leflunomide: an immunomodulatory drug for the treatment of rheumatoid arthritis and other autoimmune diseases. *Immunopharmacology* **47**, 273-289 (2000).
 37. Kalgutkar AS, *et al.* In vitro metabolism studies on the isoxazole ring scission in the anti-inflammatory agent leflunomide to its active alpha-cyanoenol metabolite A771726: mechanistic similarities with the cytochrome P450-catalyzed dehydration of aldoximes. *Drug Metab Dispos* **31**, 1240-1250 (2003).
 38. Liguori MJ, *et al.* AhR activation underlies the CYP1A autoinduction by A-998679 in rats. *Front Genet* **3**, 213 (2012).

39. Gerets HH, *et al.* Characterization of primary human hepatocytes, HepG2 cells, and HepaRG cells at the mRNA level and CYP activity in response to inducers and their predictivity for the detection of human hepatotoxins. *Cell Biol Toxicol* **28**, 69-87 (2012).
40. Westerink WM, Schoonen WG. Cytochrome P450 enzyme levels in HepG2 cells and cryopreserved primary human hepatocytes and their induction in HepG2 cells. *Toxicol In Vitro* **21**, 1581-1591 (2007).
41. Xuan J, Chen S, Ning B, Tolleson WH, Guo L. Development of HepG2-derived cells expressing cytochrome P450s for assessing metabolism-associated drug-induced liver toxicity. *Chem Biol Interact* **255**, 63-73 (2016).
42. Larigot L, Juricek L, Dairou J, Coumoul X. AhR signaling pathways and regulatory functions. *Biochim Open* **7**, 1-9 (2018).
43. Pallotta MT, Fallarino F, Matino D, Macchiarulo A, Orabona C. AhR-Mediated, Non-Genomic Modulation of IDO1 Function. *Front Immunol* **5**, 497 (2014).
44. Tanaka S, Tanaka Y, Ishiguro N, Yamanaka H, Takeuchi T. RANKL: A therapeutic target for bone destruction in rheumatoid arthritis. *Mod Rheumatol* **28**, 9-16 (2018).
45. Schett G. Autoimmunity as a trigger for structural bone damage in rheumatoid arthritis. *Mod Rheumatol* **27**, 193-197 (2017).
46. Cheloni G, *et al.* Targeting chronic myeloid leukemia stem cells with the hypoxia-inducible factor inhibitor acriflavine. *Blood* **130**, 655-665 (2017).
47. Masoud GN, Li W. HIF-1alpha pathway: role, regulation and intervention for cancer therapy. *Acta Pharm Sin B* **5**, 378-389 (2015).
48. Lee K, *et al.* Acriflavine inhibits HIF-1 dimerization, tumor growth, and vascularization. *Proc Natl Acad Sci U S A* **106**, 17910-17915 (2009).
49. Zbytek B, Peacock DL, Seagroves TN, Slominski A. Putative role of HIF transcriptional activity in melanocytes and melanoma biology. *Dermatoendocrinol* **5**, 239-251 (2013).

Reviewers' Comments:

Reviewer #1:

Remarks to the Author:

The authors have addressed most of my concerns.

Reviewer #2:

Remarks to the Author:

The authors responded to my comments point-by-point, such as animal choice, the in vivo efficient form of Leflunomide against AhR signaling, and the involved AhR pathway for Leflunomide action. I have no additional comments, and agree to the acceptance of the article for publication in Nature Communications.

Reviewer #3:

Remarks to the Author:

In this revised manuscript, the authors have satisfactorily addressed reviewers' comments and concerns.

REVIEWERS' COMMENTS:

Reviewer #1 (Remarks to the Author):

The authors have addressed most of my concerns. Our response: Thanks for the positive comments.

Reviewer #2 (Remarks to the Author):

The authors responded to my comments point-by-point, such as animal choice, the in vivo efficient form of Leflunomide against AhR signaling, and the involved AhR pathway for Leflunomide action. I have no additional comments, and agree to the acceptance of the article for publication in Nature Communications. Our response: Thanks for the positive comments.

Reviewer #3 (Remarks to the Author):

In this revised manuscript, the authors have satisfactorily addressed reviewers' comments and concerns. Our response: Thanks for the positive comments.